# Estimation of power plant $SO_2$ emissions using the HYSPLIT dispersion model and airborne observations with plume rise ensemble runs

**Tianfeng Chai**[1,2,3]**, Xinrong Ren**[1]**, Fong Ngan**[1,2,3]**, Mark Cohen**[1]**, and Alice Crawford**[1]

[1]NOAA Air Resources Laboratory (ARL), NOAA Center for Weather and Climate Prediction,
5830 University Research Court, College Park, MD 20740, USA
[2]Cooperative Institute for Satellites Earth System Studies (CISESS), University of Maryland,
College Park, MD 20740, USA
[3]Department of Atmospheric and Oceanic Science, University of Maryland, College Park, MD 20740, USA

**Correspondence:** Tianfeng Chai (tianfeng.chai@noaa.gov)

**Abstract.** The $SO_2$ emission rates from three power plants in North Carolina are estimated using the HYSPLIT Lagrangian dispersion model and aircraft measurements made on 26 March 2019. To quantify the underlying modeling uncertainties in the plume rise calculation, dispersion simulations are carried out in an ensemble using a total of 15 heat release parameters. For each heat release, the $SO_2$ emission rates are estimated using a transfer coefficient matrix (TCM) approach and compared with the Continuous Emissions Monitoring Systems (CEMS) data. An "optimal" member is first selected based on the correlation coefficient calculated for each of the six segments that delineate the plumes from the three power plants during the morning and afternoon flights. The segment influenced by the afternoon operations of Belews Creek power plant has negative correlation coefficients for all the plume rise options and is first excluded from the emission estimate here. Overestimations are found for all the segments before considering the background $SO_2$ mixing ratios. Both constant background mixing ratios and several segment-specific background values are tested in the HYSPLIT inverse modeling. The estimation results by assuming the 25th percentile observed $SO_2$ mixing ratios inside each of the five segments agree well with the CEMS data, with relative errors of 18 %, −12 %, 3 %, 93.5 %, and −4 %. After emission estimations are performed for all the plume rise runs, the lowest root mean square errors (RMSEs) between the predicted and observed mixing ratios are calculated to select a different set of optimal plume rise runs which have the lowest RMSEs. Identical plume rise runs are chosen as the optimal members for Roxboro and Belews Creek morning segments, but different members for the other segments yield smaller RMSEs than the previous correlation-based optimal members. It is also no longer necessary to exclude the Belews Creek afternoon segment that has a negative correlation between predictions and observations. The RMSE-based optimal runs result in much better agreement with the CEMS data for the previously severely overestimated segment and do not deteriorate much for the other segments, with relative errors of 18 %, −18 %, 3 %, −9 %, and 27 % for the five segments and 2 % for the Belews Creek afternoon segment. In addition, the RMSE-based optimal heat emissions appear to be more reasonable than the correlation-based values when they are significantly different for CPI Roxboro power plant.

## 1  Introduction

Both Eulerian and Lagrangian atmospheric transport models have been widely used to provide forecasts or analyses of atmospheric components for a wide range of purposes varying from emergency response to climate change predictions. However, in many applications, such as volcanic eruptions, wildfire events, accidental radionuclide releases from nuclear power plants, and climate change predictions, emissions are the most critical model input parameters but are mostly unknown and difficult to quantify. Even when emission inventories are made available through bottom-up approaches, some of the emissions are often associated with large uncertainties and systematic biases due to outdated databases, inaccurate emission factors, and invalid assumptions regarding operations, processes, and/or activities (throughput) during the bottom-up emission estimation. Therefore, various inverse modeling methods using so-called top-down approaches have been developed in order to estimate emissions by combining direct observations and the accumulated knowledge already built into atmospheric transport models. Lagrangian particle dispersion models are particularly suited to applications related to point source emission estimations because they effectively avoid calculation outside air pollutant plumes and do not have the numerical diffusion problems of most Eulerian models. Many source term estimation applications have been developed using various dispersion models and inverse modeling schemes (e.g., Stohl et al., 2012; Winiarek et al., 2012, 2014; Saunier et al., 2013; Chai et al., 2015; Bieringer et al., 2017; Hutchinson et al., 2017; Chai et al., 2018; Kim et al., 2020).

The National Oceanic and Atmospheric Administration (NOAA) Air Resources Laboratory's (ARL) HYSPLIT Lagrangian model is one of the most extensively used atmospheric transport models to simulate the atmospheric transport, dispersion, and deposition of pollutants and hazardous materials (Draxler and Hess, 1997; Stein et al., 2015). A HYSPLIT inverse system based on 4D-Var data assimilation and a transfer coefficient matrix (TCM) was developed and applied to estimate the cesium-137 source from the Fukushima nuclear accident using global air concentration measurements (Chai et al., 2015). The system was further developed to estimate the effective volcanic ash release rates as a function of time and height by assimilating satellite mass loadings and ash cloud top heights (Chai et al., 2017). More recently, the HYSPLIT-based Emissions Inverse Modeling System (HEIMS) was developed to estimate wildfire emissions from the transport and dispersion of smoke plumes captured by geostationary satellite aerosol optical depth observations (Kim et al., 2020). In another HYSPLIT inverse system study with Cross-Appalachian Tracer Experiment (CAP-TEX) data collected from six controlled releases, Chai et al. (2018) found that adding model uncertainty terms was able to improve source estimate results.

The source term estimation problem proves to be challenging because of the chaotic nature of the atmospheric flow. In addition, the observations from routine monitoring networks are typically sparse and often do not provide enough information to determine emission sources. Many field campaign studies have been carried out with airborne measurements by research aircraft in order to estimate certain air pollutant and greenhouse gas emission sources. Both traditional mass balance methods (e.g., Mays et al., 2009; Cambaliza et al., 2014; Liggio et al., 2016; Ren et al., 2018) and various inverse modeling methods which take advantage of atmospheric transport models (e.g., Karion et al., 2019; Angevine et al., 2020; Pitt et al., 2022; Lopez-Coto et al., 2022) have been applied to quantify different emissions. While many inverse modeling applications have been carried out and compared with bottom-up emission inventories, large uncertainties are still associated with top-down estimations. Karion et al. (2019) showed an intercomparison study using both the inventory scaling method and Bayesian inversion with several dispersion models and meteorological inputs for emission estimation with flight observations. They found significant variabilities (up to a factor of 3) between different models and between different days and indicated that further work was needed to evaluate and improve vertical mixing in tracer dispersion models.

To better evaluate the top-down estimates of emissions, Angevine et al. (2020) studied a power plant with Continuous Emissions Monitoring Systems (CEMS) data as the known emissions. They used a model-assisted mass balance method and examined the estimate uncertainties with an ensemble of HYSPLIT runs with different meteorological inputs and concluded with reasonably large (30 %–40 %) uncertainties for the top-down estimates of emissions. However, a constant heat release of 85 MW as the main plume rise parameter used in the Briggs formulation was specified for all the simulations. This could have caused an underestimation of the uncertainties. Gordon et al. (2018) and Akingunola et al. (2018) found that the Briggs plume rise algorithm (Briggs, 1984) significantly underestimated plume rise, in contrast to the majority of past plume rise measurement studies. A recent study by Kim et al. (2023) to estimate power plant $SO_2$ emission rates with aircraft measurements also highlighted the large uncertainties caused by the plume rise calculation when using a Gaussian footprint approach.

Fathi et al. (2021) investigated the impact of storage and release due to meteorological variability on mass balance emission rate retrieval accuracy using virtual aircraft sampling of a regional chemical transport model output. The storage-and-release events contributed to the mass balance emission estimate errors ranging from −25 % to 24 % in their tests. They recommended repeat flights around the given facility and/or time-consecutive upwind and downwind vertical profiling during the sampling period. However, inverse modeling methods using a dispersion model without assuming

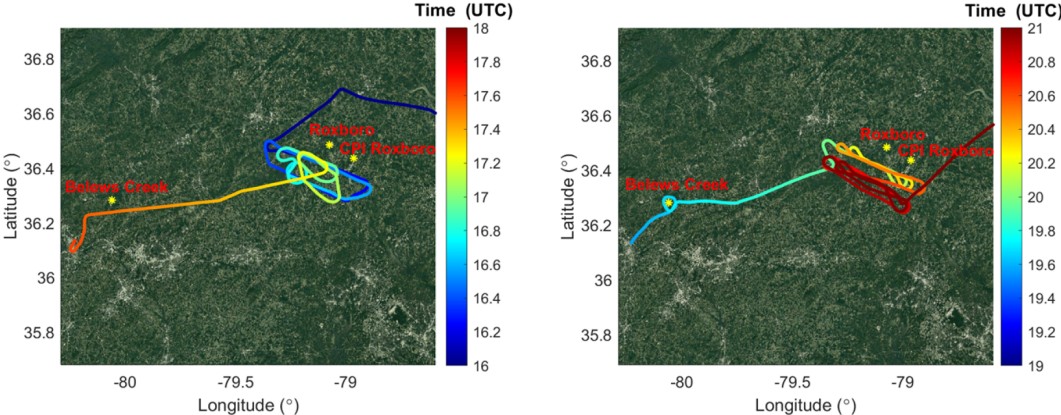

**Figure 1.** Flight tracks of the morning (left) and afternoon (right) flights on 26 March 2019 on top of the © Google Maps satellite image (retrieved in February 2023). Color represents the aircraft travel time of the day (UTC). The locations of Belews Creek, Roxboro, and CPI Roxboro power plants are also shown.

constant meteorological fields are expected to perform better than the mass balance method.

In this study, the HYSPLIT inverse modeling system is tested with flight observations collected in 2019 by the University of Maryland Cessna 402B research aircraft to estimate $SO_2$ point source emissions from three power plants in North Carolina, USA. An ensemble of model runs with a range of emission heat release parameters is used to quantify the forward model simulation uncertainties due to the plume rise calculation. The paper is organized as follows. Section 2 describes the flight observations as well as the HYSPLIT model configuration and the source term inversion method. Section 3 presents emission inversion results, and a summary is given in Sect. 4.

## 2 Methods

### 2.1 Observations

A suite of airborne measurements was collected using an instrumented small research aircraft, the University of Maryland Cessna 402B, on 26 March 2019. A morning flight started from 13:45 to 17:38 UTC and an afternoon flight lasted from 19:31 to 23:33 UTC. The flight tracks and the locations of the power plants are shown in Fig. 1. The flights were intended to sample downwind plumes originating from three coal-fired power plants in North Carolina: Roxboro (36.4833° N, 78.0731° W), CPI Roxboro (36.4350° N, 78.9619° W), and Belews Creek (36.2811° N 80.0603° W). Note that another power plant, Mayo (36.5278° N 78.8917° W), is also in the region but did not operate on the day. Measurement of $SO_2$ mixing ratios was made with a Thermo Environment model 43S pulsed fluorescence analyzer. Calibration of the $SO_2$ analyzer was conducted before and after the field study with an $SO_2$ standard that is traceable to National Institute of Standards and

Technology (NIST) reference standards. Additional measurements were also made, including aircraft locations, wind speed, wind direction, temperature, pressure, relative humidity, and mixing ratios of several other gas species, as well as some aerosol optical properties. More details related to the aircraft instruments and measurements can be found in Ren et al. (2018).

To better compare the HYSPLIT model results with the observations, the original 1 s data are averaged inside each four-dimensional (4D) HYSPLIT sampling grid box, i.e., 0.01° longitude by 0.008° latitude, 100 m in altitude, and 1 min in time in this application. It should be noted that the aircraft typically travels several three-dimensional (3D) grid boxes within a minute. The original 1 s data inside each 3D grid box are averaged separately so that multiple 1 min records would result from such a 4D averaging. For brevity, the 4D averaged data are still referred to as 1 min data hereafter.

### 2.2 HYSPLIT model

In this study, $SO_2$ plumes originating from the power plants are modeled using the HYSPLIT model (Version 5.2.0) in its particle mode in which three-dimensional (3D) Lagrangian particles released from the source location passively follow the wind field. Random velocity components based on the meteorological data are added to the mean advection velocities to simulate the dispersion process. The details of the model can be found in Draxler and Hess (1997, 1998) and Stein et al. (2015). Green et al. (2019) found that the $SO_2$ oxidation rates during the day from power plants were 0.22–0.71 % h$^{-1}$ using 13 flights from 6 February to 15 March 2015 over the eastern United States. The measurements were made during a clear-sky day on 26 March 2019 and the travel time of the measured air parcels from the stacks is less than 3 h. So it is reasonable to treat $SO_2$ as a passive tracer and ig-

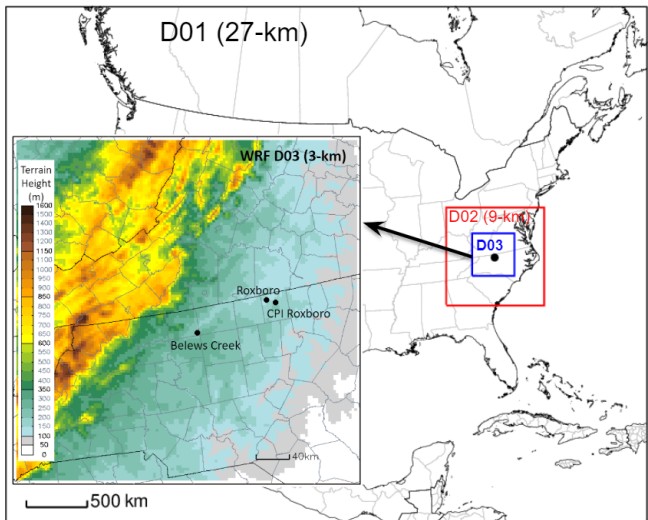

**Figure 2.** The three nested domains D03, D02, and D01 used in WRF simulations at 3, 9, and 27 km, respectively. TS1

nore its oxidation. A particle release rate of 20 000 per hour is used for all calculations. The meteorological data used to drive HYSPLIT are from the Weather Research and Forecasting (WRF; version 4.0.1) model (Powers et al., 2017). 5 The WRF model was configured for three nested domains with horizontal grid spacing of 27 km (D01), 9 km (D02), and 3 km (Fig. 2). A total of 33 vertical layers were defined with a higher resolution near the surface and 100 hPa for the model top. There were 20 layers below 850 hPa with the first 10 mid-layer height of the model at around 8 m. The simulations for D01 were initialized by using the North American Regional Reanalysis (Mesinger et al., 2006) with 32 km grid spacing and availability every 3 h. Then, the WRF results from the coarser domains provided the initial and boundary 15 conditions for the inner domains. The daily WRF runs had a 30 h duration including 6 h a spin-up period (i.e., starting at 18:00 UTC on the previous day). The physics options for the WRF simulations were the rapid radiative transfer model for radiation parameterization (Iacono et al., 2008), WSM6 for 20 microphysics (Lim and Hong, 2010), the Grell 3D ensemble for the sub-grid cloud scheme (Grell and Devenyi, 2002), the Noah land surface model (Chen and Dudhia, 2001), and the Mellor–Yamada–Nakanishi–Niino 2.5 level turbulent kinetic energy (TKE) scheme for the planetary boundary layer 25 (PBL) parameterization and its corresponding surface layer scheme (Nakanishi and Niino, 2006). In the WRF simulations, 3D grid nudging of winds is applied in the free troposphere and within the PBL. Figure 3 shows that the WRF wind speed data mostly agree well with the aircraft observa-30 tions. However, at the beginning of the afternoon flight the 1 min observations show large variations in wind direction that the 5 min WRF data cannot represent. The WRF TKE data are used to calculate the turbulent velocity variances. The ratios of the vertical to the horizontal turbulence are set

as 0.18 for both daytime and nighttime. The boundary layer 35 stability is computed from the heat and momentum fluxes from the meteorological data. The WRF mixed layer depth is directly used in the HYSPLIT model.

The dry deposition velocity of $SO_2$ is calculated using the resistance method following Wesely (1989), Chang et al. 40 (1990), and Walmsley and Wesely (1996). Note that the canopy resistance component depends upon a number of plant physiological and ground surface characteristics which are provided to the HYSPLIT model by a land use input file. The molecular weight, diffusivity ratio, and effec-45 tive Henry's law constant are specified as $64 \, \mathrm{g \, mol^{-1}}$, 1.9, and $1 \times 10^5 \, \mathrm{mol \, L^{-1} \, atm^{-1}}$, respectively. The actual Henry's constant of $1.24 \, \mathrm{mol \, L^{-1} \, atm^{-1}}$ is used to define the wet removal process for $SO_2$ as a soluble gas. The sampling grid is defined to be 0.01° longitude by 0.008° latitude and 100 m 50 in altitude from the surface to 2000 m above ground level. Mass mixing ratios are output every minute by setting the HYSPLIT parameter ICHEM to 6 to divide output mass by air density. They are later converted to volume mixing ratios by multiplying by the molecular weight ratio of air to $SO_2$. 55

## 2.3 Plume rise

The plume rise calculation in HYSPLIT is based on the Briggs formula derived from dimensional analysis for buoyancy-dominated plumes from power plant stacks (Briggs, 1969, 1984). Equation (1) shows the formulas used 60 in the HYSPLIT model for the final plume rise $\Delta H$ in different meteorological conditions following Arya (1999):

$$\Delta H = \begin{cases} 1.3 \frac{F_b}{\overline{u} u_*^2}, & \text{neutral, unstable} \\ 2.6 F_b^{1/3} \overline{u}^{-1/3} s^{-1/3}, & \text{stable}, \overline{u} > 0.5 \, \mathrm{m \, s^{-1}} \\ 5.3 F_b^{1/4} s^{-3/8}, & \text{stable}, \overline{u} \leq 0.5 \, \mathrm{m \, s^{-1}}, \end{cases} \quad (1)$$

where $F_b$ is the buoyancy flux term, $\overline{u}$ is the mean wind speed, $u_*$ is the friction velocity, and $s$ is the static stability 65 parameter as defined in Eq. (2).

$$s = \frac{g}{T_v} \frac{\partial \overline{\theta}_v}{\partial z} \quad (2)$$

Here $g$ is gravitational acceleration. $T_v$ is the moist air virtual temperature, and $\overline{\theta}_v$ is the mean virtual potential temperature. Note that the stability parameter is calculated us-70 ing the surface conditions of the meteorological data in the HYSPLIT model. A recent study by Akingunola et al. (2018) suggests a layered buoyancy approach that allows stability to change with height for the Briggs plume rise calculation. However, the layered approach is not implemented in the 75 HYSPLIT model yet. The buoyancy flux term $F_b$ is approximated by Eq. (3) (Briggs, 1969):

$$F_b = \frac{g Q_H}{\pi c_p \rho T} \approx 8.8 \times 10^{-6} \left[ \frac{m^4 \, s^{-3}}{\text{watts}} \right] Q_H[\text{watts}], \quad (3)$$

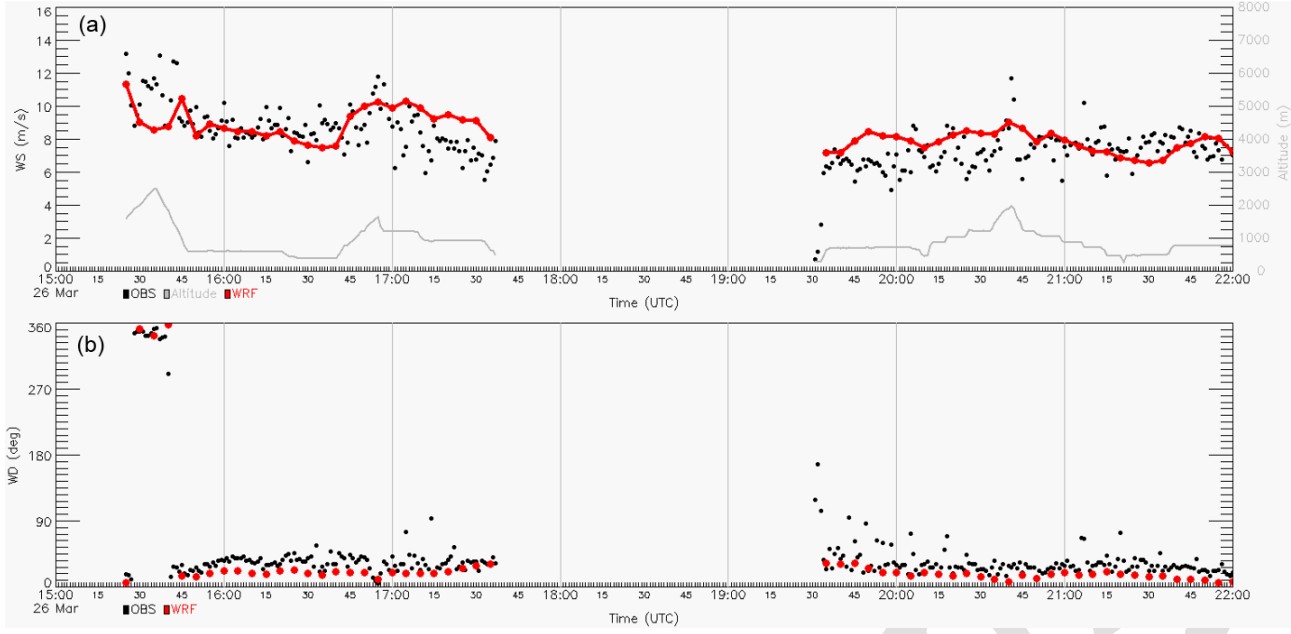

**Figure 3.** Wind speed **(a)** and wind direction **(b)** comparisons between the 1 min aircraft measurements (OBS) and 5 min WRF data along the flight. Aircraft altitudes above ground level are also shown.

where $c_p$, $\rho$, and $T$ are the specific heat at constant pressure, average density, and temperature of ambient air, respectively. $Q_H$ is the heat emission from the stack. Assuming standard atmosphere, $Q_H$ is the only user input parameter besides meteorological conditions that affects the final plume rise height $\Delta H$. It is possible to calculate $Q_H$ when the relevant parameters such as the flow rate and gas temperature at the stack exit are available. However, the exit gas temperature of the three stacks during the study period cannot be obtained. Note that even if $Q_H$ can be accurately estimated, the $\Delta H$ calculation through Eq. (1) is still subject to significant uncertainties due to some assumptions for simplification. In addition, when certain parameters are not readily available, it is preferable to assume them as unknown to allow better applicability for the source term estimation method. Thus we use a range of $Q_H$ values for plume rise height calculation to form an ensemble of dispersion runs, and the "optimal" plume rise runs that best match the observations will be selected afterwards. In detailed studies at six Tennessee Valley Authority locations over many years, it was found that heat emissions ranged from 20 to 100 MW per stack with one to nine stacks operating (Briggs, 1969). For each stack in operation, 15 heat emission values uniformly distributed from 10 to 150 MW are tested in HYSPLIT simulations. During the study period, only one stack was operating at each of the three power plants.

## 2.4 Inverse modeling method

Similar to previous HYSPLIT inverse modeling applications (e.g., Chai et al., 2015, 2017, 2018; Kim et al., 2020; Crawford et al., 2022), a transfer coefficient matrix (TCM) approach is used for the inverse modeling application. After a stack heat emission scenario is specified, 24 independent HYSPLIT Lagrangian model runs with unit hourly emissions starting from 00:00 to 23:00Z on 26 March 2019 are made at each power plant to form a TCM using the 4D averaged 1 min airborne $SO_2$ observations. A transfer coefficient at row $m$ and column $n$ of the TCM represents the source–receptor sensitivity of observation $m$ with respect to the $n$th unit-emission run from a certain source location and release hour. The unknown emissions can be solved by minimizing a cost function that integrates the differences between model predictions and observations, deviations of the final solution from the first guess (a priori), and other relevant penalty terms if needed (Daley, 1991). Following Chai et al. (2018), a cost function normalization scheme is introduced and the cost function $\mathcal{F}$ is defined as

$$\mathcal{F} = \frac{1}{2} \sum_{i=0}^{23} \sum_{j=1}^{3} \frac{(q_{ij} - q_{ij}^{b})^2}{\sigma_{ij}^2} + \frac{1}{2} \sum_{m=1}^{M} \frac{(c_m^h - c_m^o)^2}{\epsilon_m^2}$$
$$\times \frac{\sum_{m=1}^{M} \frac{1}{\epsilon_m^{b\,2}}}{\sum_{m=1}^{M} \frac{1}{\epsilon_m^2}}, \tag{4}$$

where $q_{ij}$ is the discretized source term at hour $i$ and location $j$ for which an independent HYSPLIT simulation has been run and recorded in a TCM. $q_{ij}^{b}$ is the first guess or

a priori estimate, and $\sigma_{ij}^2$ is the corresponding error variance. We assume the uncertainties of the release at each time and location are independent of each other so that only the diagonal term of the typical a priori error variance $\sigma_{ij}^2$ appears in Eq. (4). $c^{\mathrm{h}}$ and $c^{\mathrm{o}}$ denote HYSPLIT-predicted and measured mixing ratios, respectively. The observational errors $\epsilon_{\mathrm{m}}$ are assumed to be uncorrelated. Since the term $\epsilon_{\mathrm{m}}^2$ is essentially used to weight $(c_{\mathrm{m}}^{\mathrm{h}} - c_{\mathrm{m}}^{\mathrm{o}})^2$ terms, the uncertainties of the model predictions and the representative errors are included besides the observational uncertainties.

To consider $\epsilon^2$ in a simplified way, it is formulated as

$$\epsilon_{\mathrm{m}}^2 = \left(f^{\mathrm{o}} \times c_{\mathrm{m}}^{\mathrm{o}} + a^{\mathrm{o}}\right)^2 + \left(f^{\mathrm{h}} \times c_{\mathrm{m}}^{\mathrm{h}} + a^{\mathrm{h}}\right)^2. \tag{5}$$

As the additive term parameters $a^{\mathrm{o}}$ and $a^{\mathrm{h}}$ affect the $\epsilon^2$ in a similar way, the representative errors caused by comparing the measurements with the predicted concentrations averaged in a grid can be included in either $a^{\mathrm{h}}$ or $a^{\mathrm{o}}$. The multiplying factor applied to the second term in Eq. (4) is the normalization to avoid having a zero source as a spurious solution when a logarithmic metric is used in the cost function. $\epsilon_{\mathrm{m}}^{\mathrm{b}}$ represents the total uncertainties when $q_{ij}^{\mathrm{b}}$ is initially used in the model predictions. The details of the normalization can be found in Chai et al. (2018).

Chai et al. (2018) show that the logarithmic metric yields better inversion results than the original air concentration metric. In this application, the metric variable in Eq. (4) is changed to $\ln(c)$, i.e., replacing $(c_{\mathrm{m}}^{\mathrm{h}} - c_{\mathrm{m}}^{\mathrm{o}})$ with $\ln(c_{\mathrm{m}}^{\mathrm{h}}) - \ln(c_{\mathrm{m}}^{\mathrm{o}})$. In such a case, $\epsilon_{\mathrm{m}}^{\ln(c)}$ is comprised of two parts, as

$$\left(\epsilon_{\mathrm{m}}^{\ln(c)}\right)^2 = \left[\ln\left(1 + f^{\mathrm{o}} + \frac{a^{\mathrm{o}}}{c_{\mathrm{m}}^{\mathrm{o}}}\right)\right]^2$$
$$+ \left[\ln\left(1 + f^{\mathrm{h}} + \frac{a^{\mathrm{h}}}{c_{\mathrm{m}}^{\mathrm{h}}}\right)\right]^2. \tag{6}$$

Note that a constant small mixing ratio of $10^{-6}$ ppbv is added to denominators $c_{\mathrm{m}}^{\mathrm{o}}$ and $c_{\mathrm{m}}^{\mathrm{h}}$ to avoid division by zero.

## 3   Results

### 3.1   Transfer coefficient matrix

As mentioned in Sect. 2.4, a TCM approach is used in the inverse modeling. The time-varying model predictions of each independent HYSPLIT Lagrangian model run with unit hourly emissions at all the receptor time and locations are recorded as the transfer coefficients (TCs). The transfer coefficients from a set of model runs can be combined to generate a transfer coefficient matrix (TCM). Figure 4 shows a TCM with 72 columns separated into three parts representing the three power plants. Each of the 24 columns for a power plant represents a HYSPLIT run with unit hourly SO$_2$ emissions specified for a single hour on 26 March 2019. Each row indicates a 1 min 4D SO$_2$ observation with at least a nonzero

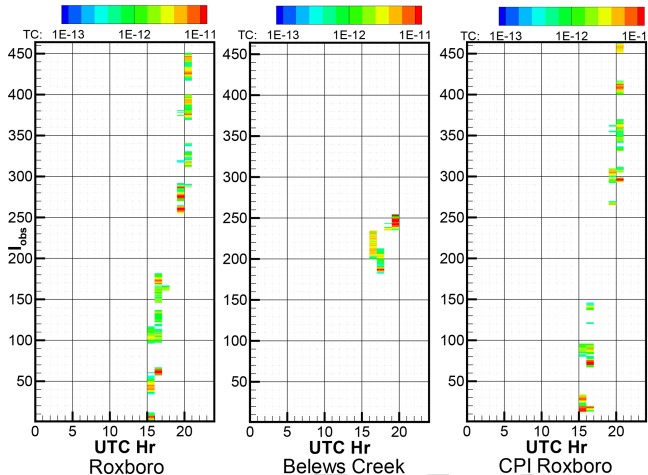

**Figure 4.** Transfer coefficients (TCs) calculated with unit hourly SO$_2$ emissions starting from 00:00 to 23:00Z on 26 March 2019 at the three power plants with $Q_{\mathrm{H}} = 50\,\mathrm{MW}$. $I_{\mathrm{obs}}$ is the index of the 1 min 4D observations ordered by their measurement time. Observations with zero transfer coefficients for all 72 HYSPLIT runs are excluded. The first 234 1 min observations belong to the morning flight, and the next 230 observations are from the afternoon flight. TC units: $\mathrm{ppbv}\,(\mathrm{kg\,h}^{-1})^{-1}$.

transfer coefficient obtained from the 72 HYSPLIT runs. The stack heat emission $Q_{\mathrm{H}} = 50\,\mathrm{MW}$ is specified for all 72 runs. A total of 464 out of 1503 1 min 4D SO$_2$ observations are affected by the three power plants during this test period, according to this set of HYSPLIT runs. Among those 464 observations, the first 234 1 min observations belong to the morning flight and the next 230 observations are from the afternoon flight. Most of the observations with zero transfer coefficients for all 72 HYSPLIT runs have low SO$_2$ mixing ratios, which are likely due to SO$_2$ background caused by minor sources other than the three power plants. Note that the background SO$_2$ mixing ratio may vary from one location to another and from one hour to the next.

Figure 4 shows that the emissions before 15:00Z or after 21:00Z of the day from any of the three power plants do not contribute to the predicted SO$_2$ plumes along the tracks of the morning or afternoon flights. Apparently the SO$_2$ emitted from the power plant stacks before 15:00Z had been transported out of the region when the aircraft measurements were made along the flight routes. Figure 1b shows that aircraft left the domain of interest at 21:00Z so that SO$_2$ emitted after 21:00Z was not sampled either. For 463 of the 464 indexed observation rows in Fig. 4, the nonzero transfer coefficients only appear in one of the three parts. That is, all observations except one are only affected by a single power plant for the current set of model runs. The only exception ($I_{\mathrm{obs}} = 369$) for the 1 min observations is influenced by both Roxboro and CPI Roxboro. When stack heat emission $Q_{\mathrm{H}} = 60\,\mathrm{MW}$ or a higher value is applied, the plumes from the three power plants are all separate without any overlap. This implies a

decoupled system in which the emission sources from the three different power plants can be solved separately. However, with lower heat emissions ($Q_H = 10$, 20, 30, 40 MW) some isolated 1 min observations may be influenced by both Roxboro and CPI Roxboro. The largest number of such observations appears when $Q_H = 10$ MW is applied to all three power plants where 6 of the 479 observations with nonzero transfer coefficients are affected by both Roxboro and CPI Roxboro. It is found that estimating the emissions from each power plant separately by ignoring the coupling effect or by removing such rare observations yields nearly identical solutions.

It is also found that the observations from the morning flights ($I_{obs} = 1$–234) and afternoon flights ($I_{obs} = 235$–464) are affected by a different set of hourly emissions. That is, none of the 72 hourly emission HYSPLIT runs contribute to both the morning flights and afternoon flights. The observations of the morning flight help to constrain the hourly emissions at 15:00Z, 16:00Z, and 17:00Z from Roxboro, the hourly emissions at 16:00Z and 17:00Z from Belews Creek, and the hourly emissions at 15:00Z and 16:00Z from CPI Roxboro, while the observations of the afternoon flight help to constrain the hourly emissions at 19:00Z and 20:00Z from Roxboro, the hourly emissions at 18:00Z and 19:00Z from Belews Creek, and the hourly emissions at 19:00Z and 20:00Z from CPI Roxboro. However, some of hourly emissions will not be well-constrained. For instance, the hourly emissions at 18:00Z from Roxboro can only be constrained by six 1 min SO$_2$ mixing ratio observations, and the hourly emission at 18:00Z from Belews Creek can only constrained by five observations. Figure 4 shows that each observation row has only one or two nonzero TC values. If there are two nonzero TC values for any observation row, they are in two consecutive columns which represent two HYSPLIT runs with hourly emissions at two consecutive hours. Instead of trying to estimate the emissions at the individual hours from each power plant, here we will only estimate the average emissions of the two or three consecutive hours that can be constrained by the morning or afternoon flights. With this decoupling approach, the cost function minimization becomes a very simplified problem.

## 3.2  Stack heat emission

As described in Sect. 2.3, when other meteorological parameters are fixed, the stack heat emission $Q_H$ becomes the single user input parameter to affect plume rise calculation with the Briggs formula being used in HYSPLIT. A total of 15 $Q_H$ values from an expected range of 10 to 150 MW are tested. For each heat emission value, 24 independent HYSPLIT Lagrangian model runs with unit hourly emissions starting from 00:00 to 23:00Z on 26 March 2019 are made at each power plant, resulting in a total of 1080 model simulations. Figure 5 shows some of the plume rise results at the three different power plant locations. Note that the plume

rise is added to the stack height listed in Table 2 for the virtual release height used in the model. The plume rise mostly goes up during the day, following the PBL development. Figure 5 also shows that the WRF PBL heights appear to be underestimated when compared with the two observation-based PBL heights estimated using the vertical potential temperature profiles. Because Roxboro and CPI Roxboro are close to each other, both the PBL heights and the plume rise results with the same $Q_H = 50$ MW are quite similar. Increasing heat emissions from $Q_H = 50$ to 100 MW at Belews Creek results in almost doubled plume rise. Conversely, a decreased heat emission from $Q_H = 50$ to 20 MW drastically reduced the plume rise.

For each heat emission value applied to a power plant, the 24 HYSPLIT simulations with unit hourly emissions can be combined together to generate the SO$_2$ plume patterns for the particular power plant. Unless there are significant hourly emission variations the correlation coefficient ($r$) between the combined plume and the observations is a good metric to evaluate the model performance without the need to estimate the emission magnitudes. Figures 6 shows the correlation coefficients between 1 min aircraft SO$_2$ observations and the unit-emission HYSPLIT simulations with different heat emissions from the three power plants. When calculating model counterparts of the observations, both horizontal nearest-neighbor and interpolation approaches are used. Note that the horizontal interpolation will increase the number of nonzero transfer coefficients in TCMs. For instance, the number of nonzero rows of the TCM in Fig. 4 increases to 570 with horizontal interpolation from the previous 464 with the nearest-neighbor option. In addition, the interpolation helps to smooth the gridded predictions. Figure 6 shows that correlation coefficients typically improve by up to 0.1 using the interpolation option. All the results presented later are with horizontal interpolation when calculating model counterparts of the observations. For the Roxboro plume, the HYSPLIT simulation with $Q_H$ between 60 and 90 MW yields a fairly good correlation between the crude predictions and observations, with $r$ equal to or better than 0.6. The best $Q_H$ for CPI Roxboro that generates better pattern matches with the observed SO$_2$ mixing ratios is probably between 40 and 90 MW, with $r$ close to 0.5. However, the simulated plume from Belews Creek only reaches reasonable correlation coefficients of $r = 0.5$ when $Q_H$ is between 120 and 140 MW. When $Q_H$ is below 80 MW, low and even negative correlation coefficients appear between the predictions and observations. This will be investigated later by separating the morning and afternoon flights.

Table 1 shows the correlation coefficients between 1 min aircraft SO$_2$ observations from the morning and afternoon flights and the model counterparts using the unit-emission HYSPLIT simulations with different heat emissions from the three power plants. For Roxboro, the HYSPLIT simulation with $Q_H = 70$ MW yields the best correlation coefficient $r = 0.68$ for the morning flight, but the best correlation

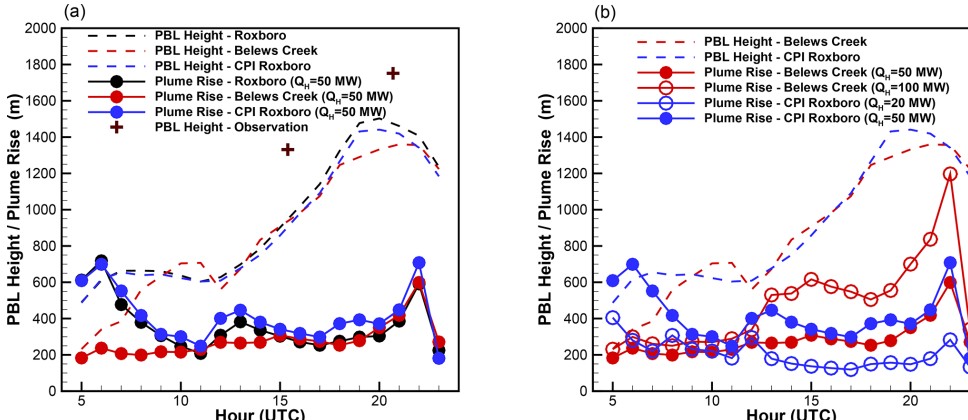

**Figure 5.** PBL heights and the final plume rise calculated with $Q_H = 50\,\mathrm{MW}$ at three different power plant locations from 05:00Z to 23:00Z on 26 March 2019 **(a)**. Two observation-based PBL heights estimated using the vertical profiles of the potential temperature from the morning and after flight measurements are also marked. Plume rises calculated with $Q_H = 100\,\mathrm{MW}$ at Belews Creek and $Q_H = 20\,\mathrm{MW}$ at CPI Roxboro are compared with those calculated with $Q_H = 50\,\mathrm{MW}$. Both PBL heights and plume rise shown are heights above ground level.

**Table 1.** Correlation coefficients between 1 min $SO_2$ observations from the morning and afternoon flights and the model counterparts using the unit-emission HYSPLIT simulations with different heat emissions from the three power plants. The highest correlation for each flight segment is highlighted with bold font, except for the Belews Creek afternoon segment, for which the highest absolute correlation coefficient is shown in italic font.

| Correlation coefficient/ | Roxboro | | Belews Creek | | CPI Roxboro | |
|---|---|---|---|---|---|---|
| Heat emission (MW) | am flight | pm flight | am flight | pm flight | am flight | pm flight |
| 10 | 0.61 | 0.46 | 0.45 | −0.62 | 0.52 | 0.05 |
| 20 | 0.60 | 0.46 | 0.49 | −0.69 | 0.60 | 0.06 |
| 30 | 0.60 | 0.44 | 0.63 | −0.64 | 0.66 | 0.09 |
| 40 | 0.62 | 0.49 | 0.63 | −0.52 | 0.72 | 0.10 |
| 50 | 0.60 | 0.55 | 0.73 | −0.28 | 0.69 | 0.19 |
| 60 | 0.67 | 0.58 | 0.83 | −0.22 | 0.72 | 0.20 |
| 70 | **0.68** | 0.58 | 0.86 | −0.28 | 0.69 | 0.22 |
| 80 | 0.64 | 0.61 | **0.87** | −0.33 | 0.74 | 0.29 |
| 90 | 0.60 | 0.62 | 0.83 | −0.58 | **0.75** | 0.35 |
| 100 | 0.55 | **0.64** | 0.82 | −0.62 | 0.64 | 0.34 |
| 110 | 0.51 | 0.60 | 0.79 | *−0.68* | 0.37 | 0.40 |
| 120 | 0.40 | 0.54 | 0.82 | −0.67 | 0.20 | 0.41 |
| 130 | 0.26 | 0.49 | 0.84 | −0.65 | 0.14 | 0.40 |
| 140 | 0.21 | 0.50 | 0.74 | −0.53 | 0.10 | **0.44** |
| 150 | 0.15 | 0.46 | 0.68 | −0.56 | 0.10 | 0.44 |

coefficient $r = 0.64$ for the afternoon is obtained when $Q_H$ is given as 100 MW. In fact, the power plant emissions had variations among the operation hours during the day. The HYSPLIT predictions of the morning and afternoon flight observations are contributed by the unit hourly emission runs from 15:00 to 17:00Z and 19:00 to 21:00Z, respectively. The CEMS $SO_2$ hourly emissions at Roxboro are 582, 345, and 360 kg h$^{-1}$ for 15:00, 16:00, and 17:00Z, respectively, and 465 and 486 kg h$^{-1}$ for 19:00 and 20:00Z, respectively. The lower average hourly emission (429 kg h$^{-1}$) contributing to the morning flight than the average hourly emission

(476 kg h$^{-1}$) contributing to the afternoon flight suggests a higher $Q_H$ for the afternoon flight than the morning flight since the emissions of $SO_2$ and heat emissions are expected to be proportional to each other for a particular stack. This agrees with the findings here; i.e., a higher optimal $Q_H$ (100 MW) is needed for a better simulation of the afternoon flight than the optimal $Q_H$ (70 MW) for the morning flight.

For the Belews Creek plume, the model results with $Q_H = 80\,\mathrm{MW}$ seem to capture the plume pattern recorded by the morning flight, with a correlation coefficient $r = 0.87$ between the 1 min observations and the HYSPLIT counter-

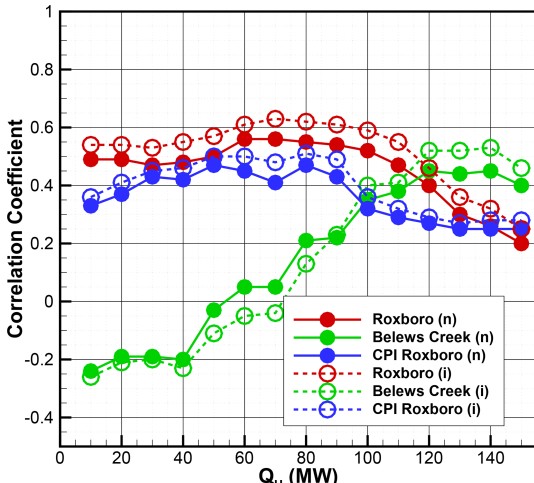

**Figure 6.** Correlation coefficients between 1 min $SO_2$ observations and the model counterparts using unit-emission HYSPLIT simulations with different heat emissions ($Q_H$) from the three power plants. When calculating model counterparts of the observations, both horizontal nearest-neighbor (n) and interpolation (i) approaches are used.

parts. However, the correlation coefficients between HYSPLIT predictions and the afternoon flight observations are all negative with all 15 $Q_H$ values. This implies problems other than plume height calculation with HYSPLIT. As shown in Fig. 3, there are large discrepancies between the WRF wind directions and the aircraft measured ones at the beginning of the afternoon flight near Belews Creek (see Fig. 1). An attempted assimilation of aircraft wind measurements using the WRF observational nudging is not quite effective to correct the wind direction biases. In addition, successful predictions of the measured $SO_2$ require wind field measurements at the upwind locations in an earlier time period, which are not available for the current case. No optimal plume rise will be selected for this segment before Sect. 3.3.3.

The HYSPLIT simulations with $Q_H = 90$ MW and $Q_H = 140$ or $150$ MW are found to correlate best with the CPI Roxboro $SO_2$ plumes measured during the morning and afternoon flights, with correlation coefficients of 0.75 and 0.44, respectively. The CEMS $SO_2$ hourly emissions at Roxboro CPI are 281 and 300 kg h$^{-1}$ for 15:00 and 16:00Z, respectively, and 316 and 295 kg h$^{-1}$ for 19:00 and 20:00Z, respectively. While the fact that the optimal $Q_H$ is higher in the afternoon corresponds well with the higher average $SO_2$ emission from the CPI Roxboro power plant at 306 kg h$^{-1}$ for 19:00–20:00Z versus 291 kg h$^{-1}$ for 15:00–16:00Z, the much lower correlation coefficient of $r = 0.44$ for the afternoon plume indicates large prediction errors even with the optimal $Q_H$ (140 or 150 MW).

Table 1 also shows that the model simulation generally performs better in the morning than in the afternoon. This is probably related to the fact that the wind directions in the afternoon are more variable than in the morning, as shown in Fig. 3. The meteorological variability may cause storage-and-release events, which make successful emission estimation more difficult to obtain, especially for the mass balance method (Fathi et al., 2021).

## 3.3 Inversion results

It has been shown that the current problem can be decoupled among the three different power plants. In addition, the $SO_2$ measurements from the power plant plumes during the morning and afternoon flights are affected by emissions of distinctive periods of 2 to 3 h. Thus six segments are considered independently. Considering the very limited number of 1 min observations to constrain the emissions at certain hours as discussed in Sect. 3.1, constant emissions are assumed for each of the six segments.

When pre-processing the observations, multiple 1 s $SO_2$ values are averaged to generate 1 min observations. The standard deviation of the multiple original 1 s observations is calculated to represent the observational uncertainty. The parameters in Eq. (5) are found using linear regression as $f^o = 0.1$ and $a^o = 0.05$ ppbv. Chai et al. (2018) found that the inversion results were not very sensitive to the observation uncertainty estimates. They also showed that setting the model uncertainty parameter to $f^m = 0.2$ yielded good results when compared with the known emission sources in the case study. Here the model uncertainty parameter of $f^m = 0.2$ is also assumed and the additive term $a^m$ is set as 0.05 ppbv, identical to $a^o$.

### 3.3.1 Zero background

Inversion estimations are first carried out without subtracting any background $SO_2$ mixing ratios from the observations. That is, the observations are assumed to originate only from the three power plant sources. Emission estimation results of the three power plants obtained by minimizing the cost functions using the morning and afternoon flights separately are listed in Table 3 with 15 different assumed heat emissions.

Based on the morning flight, the estimated Roxboro $SO_2$ emission varies from 701.5 kg h$^{-1}$ with $Q_H = 10$ MW to 473.2 kg h$^{-1}$ with $Q_H = 150$ MW. With the optimal $Q_H = 70$ MW, $SO_2$ emission is estimated as 551.9 kg h$^{-1}$, 29 % greater than the average CEMS between 15:00 and 17:00Z. Table 2 shows that the emissions at 14:00 and 15:00Z are both 582 kg h$^{-1}$, while the emissions at 16:00 and 17:00Z decrease to 345 and 360 kg h$^{-1}$ before going up again to 509 kg h$^{-1}$ at 18:00Z. The average emission from 19:00 to 20:00Z estimated based on the afternoon flight with the optimal $Q_H = 100$ MW is 520.9 kg h$^{-1}$, 9 % larger than the average CEMS value. Contrary to the morning flight, the estimated emissions are generally greater with increasing emission heat. The estimated emissions are 875.7 and 449.3 kg h$^{-1}$ with $Q_H = 150$ and 10 MW, respectively.

**Table 2.** The power plant geolocations, stack heights, and CEMS emissions (United States Environmental Protection Agency (U.S. EPA), 2022).

| Power plant name | Geolocation latitude, longitude | Stack height (m) | CEMS $SO_2$ emission ($kg h^{-1}$) | | | | | | | | | |
|---|---|---|---|---|---|---|---|---|---|---|---|---|
| | | | 13:00Z | 14:00Z | 15:00Z | 16:00Z | 17:00Z | 18:00Z | 19:00Z | 20:00Z | 21:00Z | 22:00Z |
| Roxboro | 36.483°, −79.073° | 122 | 579 | 582 | 582 | 345 | 360 | 509 | 465 | 486 | 508 | 856 |
| Belews Creek | 36.281°, −80.060° | 152 | 1349 | 1267 | 1132 | 943 | 867 | 816 | 772 | 767 | 853 | 1029 |
| CPI Roxboro | 36.435°, −78.962° | 60 | 278 | 306 | 281 | 300 | 279 | 302 | 316 | 295 | 293 | 298 |

**Table 3.** Estimation of $SO_2$ emissions from the three power plants on 26 March 2019 with 15 different assumed heat emissions and the average CEMS emissions during the specified hours. The ranges of CEMS hourly emissions for the specified hours as well as 1 h before and 1 h after the period are shown after the average CEMS emission. The relevant CEMS hourly emissions are listed in Table 2. The bold numbers are associated with the heat emissions that generate the highest correlation coefficients between observations and HYSPLIT predictions for the specific flight segments. The italic number is associated with the heat emission that generates the highest absolute correlation coefficient between observations and HYSPLIT predictions for the Belews Creek afternoon segment.

| CEMS/assumed heat emission (MW) | Roxboro | | Belews Creek | | CPI Roxboro | |
|---|---|---|---|---|---|---|
| | 15:00–17:00Z ($kg h^{-1}$) | 19:00–20:00Z ($kg h^{-1}$) | 16:00–17:00Z ($kg h^{-1}$) | 18:00–19:00Z ($kg h^{-1}$) | 15:00–16:00Z ($kg h^{-1}$) | 19:00–20:00Z ($kg h^{-1}$) |
| CEMS | 429 (345–582) | 476 (465–509) | 905 (816–1132) | 794 (767–867) | 291 (279–306) | 306 (293–316) |
| 10 | 701.5 | 449.3 | 1758.6 | 680.8 | 343.1 | 588.4 |
| 20 | 664.6 | 532.4 | 1578.7 | 512.6 | 343.8 | 590.7 |
| 30 | 636.1 | 530.3 | 1553.7 | 424.8 | 320.5 | 572.2 |
| 40 | 806.8 | 740.4 | 3735.0 | 339.0 | 557.5 | 503.5 |
| 50 | 617.3 | 491.5 | 1547.5 | 339.0 | 398.5 | 478.9 |
| 60 | 611.1 | 506.5 | 1475.0 | 283.8 | 402.5 | 457.4 |
| 70 | **551.9** | 529.3 | 1445.9 | 298.9 | 464.8 | 475.1 |
| 80 | 538.9 | 488.6 | **1417.3** | 393.0 | 504.5 | 429.1 |
| 90 | 520.9 | 485.7 | 1451.7 | 368.7 | **712.8** | 412.3 |
| 100 | 514.7 | **520.9** | 1411.6 | 485.7 | 1095.3 | 416.4 |
| 110 | 525.0 | 515.7 | 1550.6 | *697.3* | 1372.7 | 413.9 |
| 120 | 512.6 | 564.2 | 1406.0 | 1027.4 | 2186.5 | 361.4 |
| 130 | 521.9 | 593.1 | 1716.9 | 707.1 | 2627.7 | 357.1 |
| 140 | 474.2 | 789.2 | 1815.7 | 818.1 | 5214.3 | **384.4** |
| 150 | 473.2 | 875.7 | 1986.5 | 926.0 | 4261.5 | 401.7 |

Using the morning flight observations, Belews Creek $SO_2$ emissions between 16:00 and 17:00Z are overestimated with all 15 heat emissions. With the optimal $Q_H = 80$ MW, the estimated emission is $1417.3\,kg h^{-1}$. Although this is 57 % larger than the average CEMS emission, it is better than the estimates with other $Q_H$ values except $Q_H = 100$ and 120 MW, which yield slightly lower emissions (1417.3 TS2 and $1406.0\,kg h^{-1}$).

It is also noted that estimated Belews Creek $SO_2$ emissions using the afternoon flight observations are mostly within a factor of 2 when compared with the average hourly CEMS emissions, while significant negative correlations are found between the observations and the model predictions. The worst underestimations when $Q_H = 40$–70 MW are associated with lower absolute correlations ($|r| < 0.3$). At $Q_H = 110$ MW when the most extreme anticorrelation ($r = -0.68$) occurs, the estimated $SO_2$ emission of $697.3\,kg h^{-1}$ is very close to the average hourly CEMS emission of $794\,kg h^{-1}$ between 18:00 and 19:00Z. The inverse corre-

lation is caused by the plume misplacement mostly due to wind direction error. The high absolute correlation indicates that the model probably predicts the mixing ratio gradient relatively well but misplaces the plume relative to the actual plume. Since the model predicts higher mixing ratios when observation values are low but predicts lower mixing ratios when observation values are higher, neither lower nor higher emissions would improve the agreement between the predictions and observations. Thus, no significant biases arise from such cases. Nonetheless, the negative correlations between the model and observations indicate model deficiencies and require special attention.

The CPI Roxboro emission estimates based on the morning and afternoon flights with the optimal $Q_H$ values (90 and 140 MW) are 712.8 and $384.4\,kg h^{-1}$, respectively. They are overestimated over the CEMS by 145 % and 26 %. The CPI Roxboro emission estimates based on the morning flight increase significantly when $Q_H$ is above 100 MW. Overestimations of the $SO_2$ emissions by factors of 18 and 15 are

**Table 4.** Number of 1 min $SO_2$ observations and some statistics of the $SO_2$ mixing ratios. There is overlap between Roxboro and CPI Roxboro segments since some observations are affected by both power plants.

| Number of $SO_2$ observations /$SO_2$ mixing ratio (ppbv) | All observations | Roxboro | | Belews Creek | | CPI Roxboro | | Missed observations |
|---|---|---|---|---|---|---|---|---|
| | | morning | afternoon | morning | afternoon | morning | afternoon | |
| Number of observations | 1503 | 192 | 186 | 55 | 23 | 118 | 153 | 810 |
| Minimum | 0.001 | 0.002 | 0.001 | 0.147 | 0.011 | 0.031 | 0.025 | 0.001 |
| 5th percentile | 0.032 | 0.045 | 0.032 | 0.332 | 0.011 | 0.051 | 0.045 | 0.026 |
| 10th percentile | 0.058 | 0.066 | 0.066 | 0.589 | 0.019 | 0.083 | 0.094 | 0.046 |
| 25th percentile | 0.136 | 0.128 | 0.166 | 1.041 | 0.038 | 0.175 | 0.209 | 0.114 |
| Median | 0.257 | 0.307 | 0.398 | 2.002 | 0.500 | 0.297 | 0.351 | 0.199 |
| 75th percentile | 0.465 | 0.586 | 0.611 | 2.826 | 0.665 | 0.493 | 0.538 | 0.317 |
| Maximum | 7.249 | 2.862 | 1.626 | 7.249 | 3.780 | 1.578 | 1.246 | 1.721 |

found with $Q_H$ set as 140 and 150 MW, respectively. Table 1 shows that the two heat emissions yield correlation coefficients of 0.10, a significant drop from $r = 0.75$ when $Q_H$ is assumed to be 90 MW. Although the highest correlation coefficient between observations and unit-emission HYSPLIT predictions for a specific flight segment may not produce the best emission estimates, a low correlation coefficient typically indicates modeling deficiencies very effectively.

### 3.3.2 $SO_2$ background

With zero background $SO_2$ mixing ratios, the emission estimates based on the optimal heat emission are all greater than the CEMS emissions. This indicates that it is necessary to consider the $SO_2$ background mixing ratios. The HYSPLIT simulated mixing ratios are actually the enhancements over the background mixing ratios. As shown in Table 4, there are 810 1 min observations, which is more than half of the 1503 1 min $SO_2$ observations not residing in any of the HYSPLIT simulated plumes originating from the three power plants with any of the 15 heat emissions. It has to be noted that the flight patterns could have been better constructed. Sampling upwind as well as downwind or in closed-shape flight patterns which enclose the sources (see, e.g., Ryoo et al., 2019, Fathi et al., 2021, and Kim et al., 2023) would have significantly helped in the estimation of the $SO_2$ background mixing ratios.

At first, the median value of the missed $SO_2$ observations (0.199 ppbv) is assumed to be the background $SO_2$ mixing ratio. This value is subtracted from all the observations unless the values are below this background value, where the observations are set as zero. Using the adjusted observations, the emission estimation results are listed in Table 5. Compared to the estimates with zero background mixing ratios, the estimated emissions are all reduced, as expected. The Roxboro emissions are estimated to be 436 kg h$^{-1}$ for 15:00–17:00Z and 403.3 kg h$^{-1}$ for 19:00–20:00Z. The morning segment estimate agrees much better with the CEMS than the previous estimate without considering the background $SO_2$ mixing ra-

tios. The estimated Belews Creek emission of 1259.7 kg h$^{-1}$ is significantly improved as well. The CPI Roxboro emission during the 15:00–16:00Z period is overestimated by 89 %, but it is not as severe as the previous 145 % overestimation. The estimated CPI Roxboro emission for the 19:00–20:00Z period is within 4 % of the CEMS value.

Table 4 shows the statistical distribution values of the six different segments, i.e., the morning and afternoon plumes from the three power plants. The highest 1 min $SO_2$ mixing ratio of 7.249 ppbv is inside the Belews Creek plume measured during the morning flight. The observed $SO_2$ mixing ratios inside the Belews Creek plumes are much higher than those from the other plumes. It is beneficial to assume different background values for the six different segments of the observations. The minimum, the 5th percentile, the 10th percentile, and the 25th percentile mixing ratios of the morning and afternoon observations inside the plumes from three different power plants are assumed to be segment-specific background mixing ratios. After subtracting the assumed background values from the observations, the emission estimation results are listed in Table 5. The estimated emissions decrease with increasing background values. With the segment-specific 25th percentile as the background, the Belews Creek emission estimation of 929.7 kg h$^{-1}$ is within 3 % of the CEMS values, and the other estimates are comparable to the results by assuming a constant background mixing ratio of 0.199 ppbv.

### 3.3.3 Root mean square errors (RMSEs)

Up to now, the best heat emission parameters have been selected based on the correlation coefficients between the observations and predicted counterparts for each segment of the observations after an ensemble of HYSPLIT runs with 15 different heat emissions. This can be performed before the emissions are estimated since the correlation coefficients are not affected by the magnitudes of the emissions when emissions for each segment are assumed to be constant. After the

**Table 5.** Estimation of $SO_2$ emissions from the three power plants on 26 March 2019 with different background mixing ratios. The "optimal" heat emission that generates the highest correlation coefficient between observations and unit-emission HYSPLIT predictions for the specific flight segment is assumed. Complete emission estimates with all heat emissions and different background mixing ratios are listed in Tables 3, A1–A4, and 6. The average CEMS emissions during the specified hours are listed for reference. The relevant CEMS hourly emissions are listed in Table 2. The segment-specific statistical distribution values are listed in Table 4.

| CEMS/background | Roxboro | | Belews Creek | CPI Roxboro | |
|---|---|---|---|---|---|
| $SO_2$ mixing ratios | 15:00–17:00Z $(\mathrm{kg\,h^{-1}})$ | 19:00–20:00Z $(\mathrm{kg\,h^{-1}})$ | 16:00–17:00Z $(\mathrm{kg\,h^{-1}})$ | 15:00–16:00Z $(\mathrm{kg\,h^{-1}})$ | 19:00–20:00Z $(\mathrm{kg\,h^{-1}})$ |
| CEMS | 429 | 476 | 905 | 291 | 306 |
| | 551.9 | 520.9 | 1417.3 | 712.8 | 384.4 |
| 0.199 ppbv | 436.0 | 403.3 | 1259.7 | 549.7 | 294.7 |
| Minimum, segment-specific min | 550.8 | 517.8 | 1316.3 | 684.9 | 371.6 |
| 5th percentile, segment-specific | 518.8 | 502.5 | 1210.3 | 659.4 | 359.9 |
| 10th percentile, segment-specific | 503.5 | 481.8 | 1067.2 | 628.5 | 335.6 |
| 25th percentile, segment-specific | 461.1 | 418.9 | 929.7 | 563.1 | 294.1 |

emission magnitudes are estimated, model performance can be evaluated using other statistical metrics.

The correlation-based emission estimations using all 15 different heat emission parameters by assuming the segment-specific 25th percentile observation to be the background mixing ratios are listed in Table 6. The root mean square errors (RMSEs) of the HYSPLIT predicted morning and afternoon plumes from the three power plants with all the plume rise ensemble runs are listed in Table 7. The optimal heat emissions that yield the best correlation coefficients also result in the smallest RMSEs for two segments, the morning plumes from Roxboro and Belews Creek. The afternoon plume from Roxboro predicted with $Q_H = 90\,\mathrm{MW}$ and the estimated emission of $389.9\,\mathrm{kg\,h^{-1}}$ has the smallest RMSE of 0.428 ppbv. The emission is underestimated by 18 %. However, for both the morning and afternoon plumes from CPI Roxboro, optimal heat emissions associated with the highest correlation coefficients are quite different from the heat emissions that produce the smallest RMSEs. If the model runs associated with the smallest RMSEs are selected, the estimated CPI Roxboro $SO_2$ emissions are $265.1\,\mathrm{kg\,h^{-1}}$ for 15:00–16:00Z and $389.1\,\mathrm{kg\,h^{-1}}$ for 19:00–20:00Z, which are 9 % underestimated and 27 % overestimated compared to CEMS. While the 19:00–20:00Z emission is worse than the result based on the best correlation, the 15:00–16:00Z emission estimation is much closer to the CEMS than the correlation-based result, which is 94 % overestimated. For the plume from Belews Creek observed during the afternoon flight, $Q_H = 140\,\mathrm{MW}$ yields the lowest RMSE of 1.874 ppbv, which is more than 3 times the median $SO_2$ observation in the segment. The lowest RMSE of 0.859 ppbv for the Belews Creek morning segment is smaller than the 25th percentile value of the observation (1.041 ppbv). For the other four segments, the best RMSEs are slightly larger than the median of the observations. This indicates the poor performance of the Belews Creek afternoon model simula-

tion. However, the emission inversion with $Q_H = 140\,\mathrm{MW}$ still yields a very good estimate of $811.6\,\mathrm{kg\,h^{-1}}$, which is only 2 % overestimated.

Figure 7 shows the comparison of both the RMSE-based and correlation-based optimal predictions with the morning and afternoon flight observations in the HYSPLIT predicted plumes from the three power plants. Identical results are obtained using the smallest RMSE and the highest correlation coefficient for the morning segments from Roxboro and Belews. For both cases, the predicted $SO_2$ mixing ratios agree well with the observations. Note that here the $SO_2$ predictions include both the predicted $SO_2$ enhancement with the estimated emissions and the assumed segment-specific background values, which are chosen as the 25th percentile observations inside the particular plumes. For the other cases, the RMSE-based predictions tend to produce lower mixing ratios for the observed high $SO_2$ values. Thus the linear regression lines for the RMSE-based predictions tend to have flatter slopes. However, the RMSE-based emission can still be larger, such as the CPI Roxboro afternoon case. The scatter plot for the Belews Creek afternoon case clearly shows anticorrelation as indicated by the negative correlation coefficients listed in Table 1. This is caused by plume misplacement due to wind direction errors. Although the predicted high and low mixing ratios are opposite to the observations, the minimization of the cost function defined by Eq. (4) is still capable of reaching an estimate close to the actual emission rate. The observations appear to have a good representation of the mixing ratio distribution for the plume at the distance from the source. Even if the model misplaced the plume location, predicted mixing ratios that have a similar distribution of the low and high values still have the minimal cost function. That is, the inverse modeling method is not very sensitive to plume misplacement. If $Q_H = 110\,\mathrm{MW}$ that generates the highest negative correlation of $-0.68$ is chosen as the optimal plume rise parameter, the estimated

**Table 6.** Estimated $SO_2$ emissions from the three power plants on 26 March 2019 with 15 different assumed heat emissions and the average CEMS emissions during the specified hours. The segment-specific 25th percentile observations are assumed to be the background $SO_2$ mixing ratios and have been subtracted from the observations for emission inversion. The ranges of CEMS hourly emissions for the specified hours as well as 1 h before and 1 h after the period are shown after the average CEMS emission. The relevant CEMS hourly emissions are listed in Table 2. The bold numbers are associated with the heat emissions which generate the highest correlation coefficients between observations and HYSPLIT predictions for the specific flight segments. The underlined numbers are associated with the smallest RMSEs listed in Table 7. The italic number is associated with the heat emission that generates the highest absolute correlation coefficient between observations and HYSPLIT predictions for the Belews Creek afternoon segment.

| CEMS/assumed | Roxboro | | Belews Creek | | CPI Roxboro | |
|---|---|---|---|---|---|---|
| heat emission (MW) | 15:00–17:00Z (kg h$^{-1}$) | 19:00–20:00Z (kg h$^{-1}$) | 16:00–17:00Z (kg h$^{-1}$) | 18:00–19:00Z (kg h$^{-1}$) | 15:00–16:00Z (kg h$^{-1}$) | 19:00–20:00Z (kg h$^{-1}$) |
| CEMS | 429 (345–582) | 476 (465–509) | 905 (816–1132) | 794 (767–867) | 291 (279–306) | 306 (293–316) |
| 10 | 609.9 | 365.7 | 1290.2 | 701.5 | 292.4 | 481.8 |
| 20 | 576.7 | 425.7 | 1165.2 | 512.6 | 290.6 | 495.5 |
| 30 | 545.4 | 427.4 | 1121.8 | 418.9 | 265.1 | 483.8 |
| 40 | 674.0 | 568.7 | 2606.8 | 332.9 | 479.9 | 420.6 |
| 50 | 531.4 | 394.6 | 1077.9 | 330.3 | 311.7 | 389.1 |
| 60 | 517.8 | 403.3 | 975.4 | 272.6 | 320.5 | 373.8 |
| 70 | **461.1** | 427.4 | 935.3 | 287.8 | 358.5 | 395.4 |
| 80 | 445.7 | 399.3 | **929.7** | 378.4 | 394.6 | 352.8 |
| 90 | 436.9 | 389.9 | 948.5 | 362.1 | **563.1** | 336.3 |
| 100 | 434.3 | **418.9** | 913.2 | 480.9 | 895.1 | 353.5 |
| 110 | 436.9 | 424.0 | 995.1 | *709.9* | 1162.9 | 324.4 |
| 120 | 434.3 | 467.6 | 922.3 | 1062.9 | 1947.2 | 282.6 |
| 130 | 454.7 | 490.6 | 1097.4 | 731.5 | 2509.7 | 282.1 |
| 140 | 408.2 | 634.8 | 1249.6 | 811.6 | 5373.0 | **294.1** |
| 150 | 416.4 | 730.1 | 1445.9 | 924.2 | 3176.9 | 311.1 |

**Table 7.** RMSEs of the $SO_2$ mixing ratios of morning and afternoon plumes from three power plants calculated using the estimated $SO_2$ emissions from the three power plants with 15 different assumed heat emissions listed in Table 6. The italic number is associated with the heat emission that generates the highest absolute correlation coefficient between observations and HYSPLIT predictions for the Belews Creek afternoon segment. Bold numbers are associated with the heat emissions which generate the highest correlation coefficients between observations and unit-emission HYSPLIT predictions for the specific flight segments. The underlined numbers indicate the smallest RMSEs of each segment.

| SO$_2$ RMSE (ppbv)/ | Roxboro | | Belews Creek | | CPI Roxboro | |
|---|---|---|---|---|---|---|
| Assumed heat emission (MW) | morning | afternoon | morning | afternoon | morning | afternoon |
| 10 | 0.635 | 0.429 | 1.409 | 2.590 | 0.612 | 0.538 |
| 20 | 0.640 | 0.469 | 1.368 | 2.140 | 0.525 | 0.564 |
| 30 | 0.635 | 0.486 | 1.242 | 2.079 | 0.438 | 0.566 |
| 40 | 0.684 | 0.681 | 2.706 | 2.212 | 1.299 | 0.984 |
| 50 | 0.539 | 0.442 | 1.106 | 2.520 | 0.509 | 0.470 |
| 60 | 0.444 | 0.478 | 0.916 | 2.681 | 0.464 | 0.527 |
| 70 | **0.434** | 0.476 | 0.918 | 2.412 | 0.470 | 0.559 |
| 80 | 0.471 | 0.431 | **0.859** | 2.222 | 0.451 | 0.522 |
| 90 | 0.455 | 0.428 | 1.031 | 1.916 | **0.496** | 0.527 |
| 100 | 0.481 | **0.456** | 1.040 | 1.905 | 0.586 | 0.630 |
| 110 | 0.511 | 0.488 | 1.290 | *2.334* | 0.871 | 0.630 |
| 120 | 0.563 | 0.589 | 1.299 | 2.879 | 1.777 | 0.699 |
| 130 | 0.725 | 0.652 | 1.362 | 2.120 | 2.679 | 0.665 |
| 140 | 0.766 | 0.838 | 1.590 | 1.874 | 4.701 | **0.553** |
| 150 | 0.893 | 0.866 | 1.630 | 1.903 | 2.956 | 0.563 |

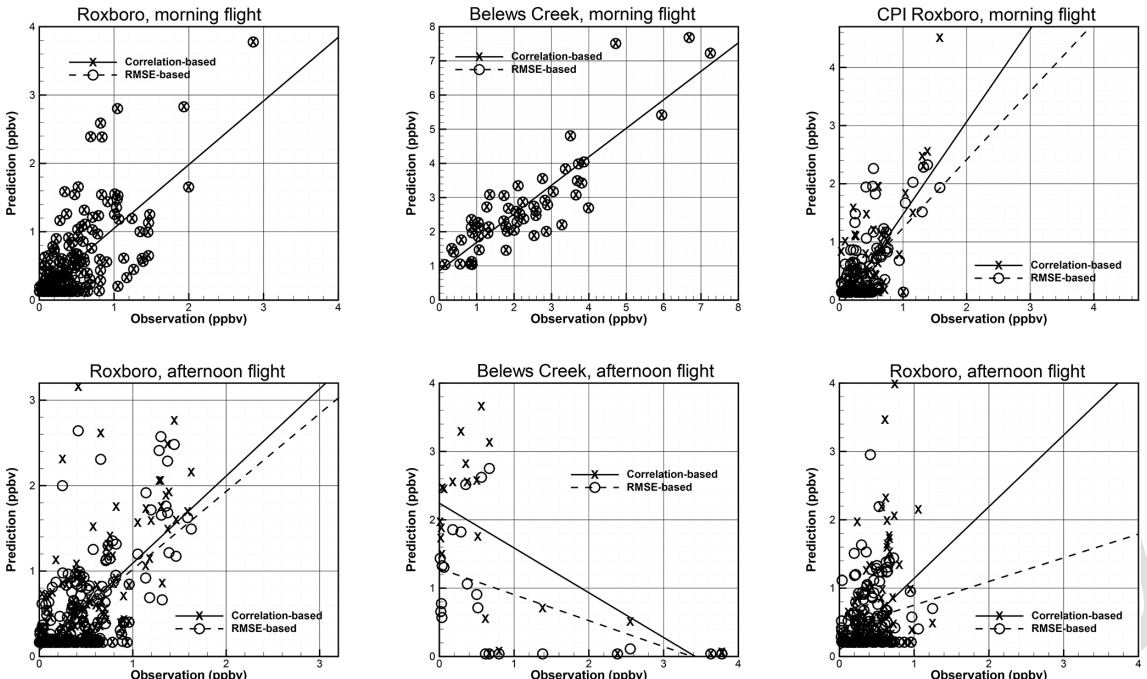

**Figure 7.** Comparison of the correlation-based and the RMSE-based "optimal" predictions with the morning and afternoon flight observations in the HYSPLIT predicted $SO_2$ plumes from the three power plants. The correlation-based predictions are with the $Q_H$ values which generate the highest correlation coefficients listed in Table 1. The highest absolute correlation coefficient is selected for the Belews Creek afternoon flight case. The RMSE-based predictions are associated with the cases which generate the smallest RMSEs listed in Table 7. The linear regression lines are shown for both the correlation-based and the RMSE-based predictions with the observations.

emission for Belews Creek during the 18:00–19:00Z period is $709.9\,kg\,h^{-1}$, which is only 11 % lower than the CEMS value of $794\,kg\,h^{-1}$. It might still be possible to have reasonable emission inversion results even when plumes are mis-
5  placed by the model.

    Figure 8 shows the optimal predictions based on the highest correlation coefficients and minimal RMSEs at 800 m a.g.l. at 17:00 and 19:00Z. Continuous vertical profiles along the flight track, or "curtain" plots, of the correlation-
10  based and RMSE-based optimal predictions are shown in Fig. 9 and enlarged in Figs. 10 and A1–A9. For the morning flight, the optimal predictions of the Roxboro and Belews Creek plumes based on the highest correlation coefficient and minimal RMSE are identical. The prediction results agree
15  well with the observed plume placement and width, as well as the mixing ratios. On the other hand, for the CPI Roxboro morning plume, the RMSE-based optimal prediction with $Q_H = 30\,MW$ is quite different from the correlation-based optimal prediction with $Q_H = 90\,MW$. The center
20  of the RMSE-based plume is at a lower altitude than the correlation-based plume (Figs. A1, A2, and A4). The lower-placed plume is also associated with lower mixing ratios that match the observations better. Figure 8c shows a wider CPI Roxboro plume of the RMSE-based result than the
25  correlation-based result in Fig. 8a. The larger extent of the RMSE-based CPI Roxboro plume results in an extra appear-

ance of the plume under the flight track in the curtain plot (Fig. A3).

    For the Roxboro plume captured during the after-
noon flight, the correlation-based optimal prediction with  30
$Q_H = 100\,MW$ and $SO_2$ emissions of $418.9\,kg\,h^{-1}$ shows very similar spatial structures and mixing ratios as the RMSE-based optimal prediction with $Q_H = 90\,MW$ and $389.9\,kg\,h^{-1}$. Figures 8, 10, and A7–A9 show little difference between the two, and both agree well with the 1 min  35
aircraft observations. Figure 10 shows that both predictions underestimated the observed peak values along the flight, but the peak location and the width of the Roxboro plumes match well between the predictions and observations. Note that the $SO_2$ emissions are underestimated by both of the optimal se-  40
lections for this segment.

    As shown in Table 1, strong anticorrelation is found between predicted and observed $SO_2$ mixing ratios of the Belews Creek afternoon plume. The prediction with $Q_H = 110\,MW$ that has highest absolute correlation coef-  45
ficient is selected here as the correlation-based solution. Figure A6 shows that it is not very different from the RMSE-based result with $Q_H = 140\,MW$. Both cases clearly misplaced the first transect of the plume and predicted wider transects than the observations. It is found that the  50
second transect shown in Fig. A6 is well-predicted with $Q_H = 110\,MW$.

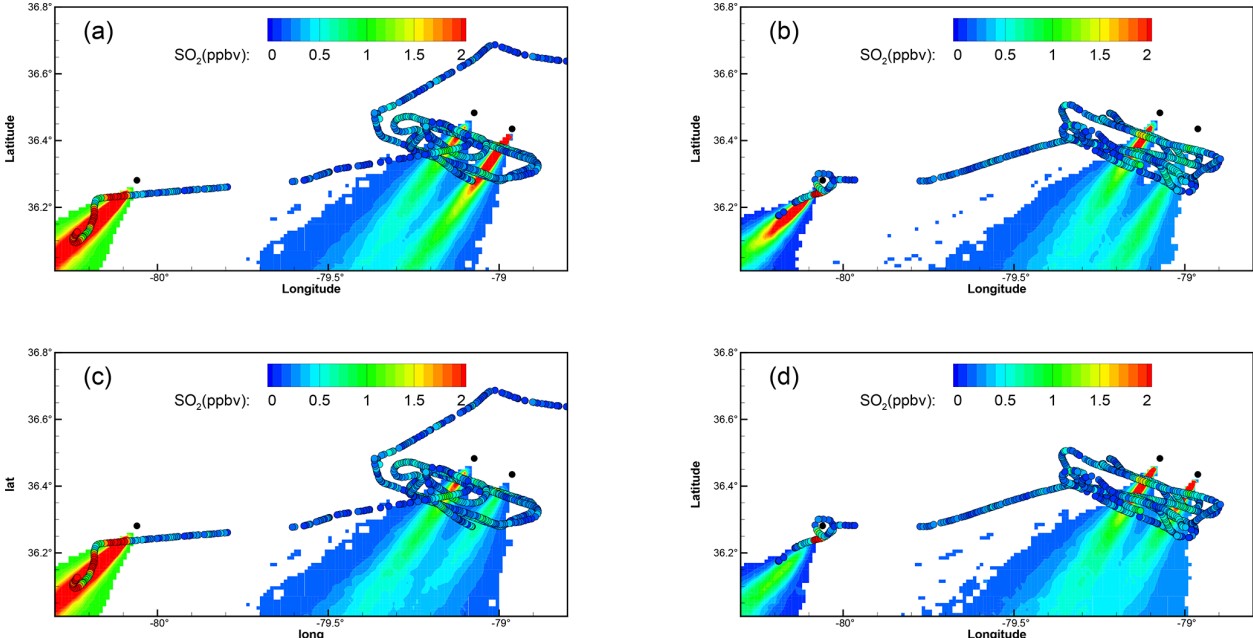

**Figure 8.** Comparison of the correlation-based **(a, b)** and the RMSE-based **(c, d)** "optimal" predictions at 800 m a.g.l. at 17:00Z **(a, c)** and 19:00Z **(b, d)**. The morning **(a, c)** and afternoon **(b, d)** 1 min observations are overlaid as circles. Color indicates the $SO_2$ values for both predictions and observations. The three power plants are marked with solid black circles.

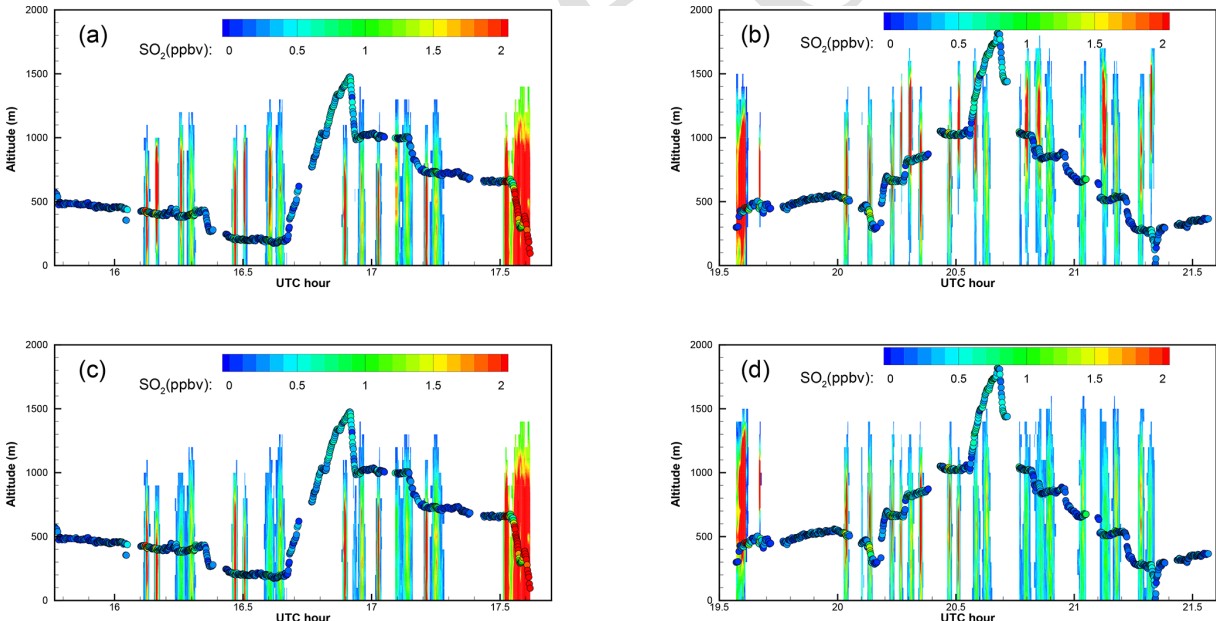

**Figure 9.** "Curtain" plots of the correlation-based **(a, b)** and the RMSE-based **(c, d)** "optimal" predictions. In the curtain plots, continuous vertical profiles along the flight track are shown following the observation time. The morning **(a, c)** and afternoon **(b, d)** 1 min observations are overlaid as circles. Color indicates the $SO_2$ values for both predictions and observations.

For the CPI Roxboro plume observed during the afternoon flight, the correlation-based optimal prediction with $Q_H = 140$ MW and the RMSE-based optimal prediction with $Q_H = 50$ MW appear drastically different in Figs. 8, 10, and A7–A9, as expected. Figures 10 and A7–A9 show

that the RMSE-based optimal prediction has wider plume transects and has them placed at lower altitudes than the correlation-based results. The predicted mixing ratios match the observations much better than the correlation-based results, although the estimated emission of $389.1 \, \mathrm{kg \, h^{-1}}$

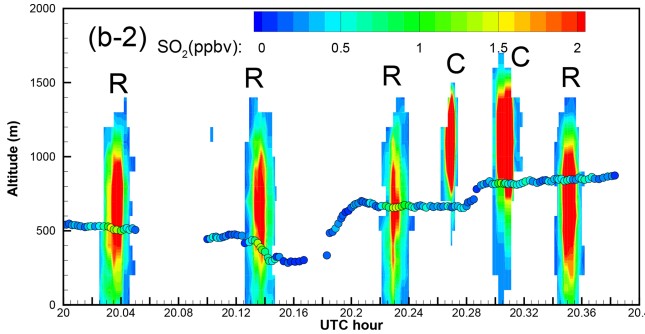

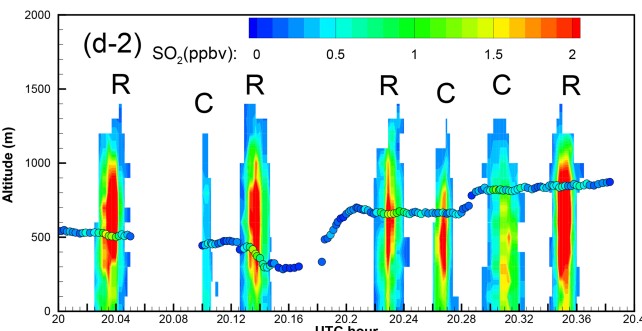

**Figure 10.** Enlarged "curtain" plots of the correlation-based **(b)** and the RMSE-based **(d)** "optimal" predictions in Fig. 9 (Part 2). Five portions of the afternoon curtain plot are enlarged, with the earliest period as Part 1 and the latest one as Part 5. In the curtain plots, continuous vertical profiles along the flight track are shown following the observation time. The afternoon flight 1 min observations are overlaid as circles. Color indicates the $SO_2$ values for both predictions and observations. Predicted plumes from Roxboro, Belews Creek, and CPI Roxboro are indicated with letters R, B, and C, respectively.

is not closer to the CEMS emission of $306\,kg\,h^{-1}$ than the correlation-based estimation of $294.1\,kg\,h^{-1}$. In addition, Fig. 10 shows that the RMSE-based solution captures an observed narrow CPI Roxboro plume transect that the correlation-based solution fails to reproduce. The results here indicate the need to have more observations at different altitudes in future flight planning.

## 4 Summary and discussion

An ensemble of HYSPLIT runs with various heat release parameters for the Briggs plume rise algorithm is made to estimate $SO_2$ emissions from three power plants. Using a TCM approach for the inverse modeling, independent HYSPLIT Lagrangian model runs with unit hourly emissions are carried out for each heat release value. The $SO_2$ emissions from the three power plants during the morning and afternoon flight periods on 26 March 2019 are estimated separately through six different segments.

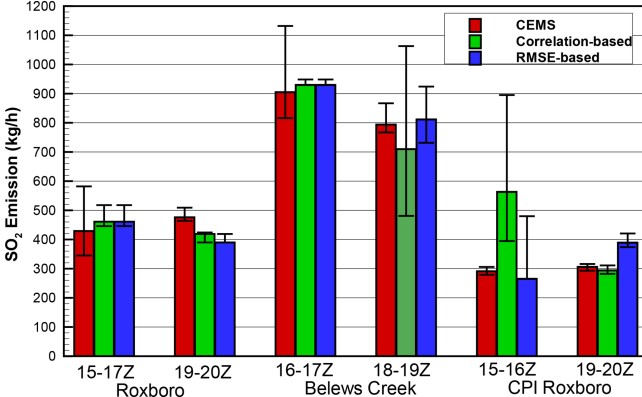

**Figure 11.** The CEMS and estimated $SO_2$ emissions from the three power plants on 26 March 2019 during the specified hours. Error bars of CEMS emissions indicate the ranges of hourly emissions for the specified hours as well as 1 h before and 1 h after. Correlation-based and RMSE-based estimates are the inversion results using the "optimal" heat emission that generates the highest correlation coefficient and the smallest RMSE between observations and the HYSPLIT predications for the specific flight segment, respectively. The correlation-based Belews Creek afternoon segment is based on the highest absolute correlation coefficient. Error bars of the estimated $SO_2$ emissions show the ranges of the results using 10 MW above and below the optimal heat emissions.

Initially the "optimal" plume rise runs are selected based on the highest correlation coefficients between predictions and observations. A segment with negative correlations is excluded. It is found that the $SO_2$ emissions are overestimated for all the remaining segments if background mixing ratios are not considered. Several different assumptions of background values are then tested. Assuming the 25th percentile observed $SO_2$ mixing ratio inside each segment to be the background $SO_2$ mixing ratios yields good emission estimates, with relative errors of 18 %, −12 %, 3 %, 93.5 %, and −4 % when compared with the CEMS data (see Fig. 11). Note that the ranges of the inverted emissions with 10 MW above and below the optimal heat emissions are used to indicate the sensitivities of the results to the heat emissions. While the differences between the emission estimates and the known CEMS data provide some confidence to the results, quantification of the uncertainties associated with the method probably requires further investigation in the future.

Using the same segment-specific $SO_2$ background assumption, optimal plume rise runs are later selected to have the smallest RMSEs between the predicted and observed mixing ratios. The previously excluded segment that has negative correlation coefficients between predictions and observations is also included in the emission inversion. While identical plume rise runs are chosen as the optimal members for Roxboro and Belews Creek morning segments, different runs are selected for the other three segments than the previous correlation-based results. In addition, emission in-

version for the previously excluded segment that has negative correlation coefficients between predictions and observations is also carried out. The relative errors are 18 %, −18 %, 3 %, −9 %, and 27 % for the five segments and 2 % for the Belews Creek afternoon segment. Figure 11 shows that the RMSE-based estimate of $SO_2$ emissions from CPI Roxboro at 15:00–16:00Z agrees much better with the CEMS data than the correlation-based estimate does. The RMSE-based $SO_2$ emission estimates of Roxboro at 19:00–20:00Z and CPI Roxboro at 19:00–20:00Z appear to deteriorate slightly. However, the associated HYSPLIT predictions show better agreement with the observations than the correlation-based optimal runs because of their smaller RMSEs.

While the stack exit gas temperature data are not available for this study, a single constant stack exit temperature is provided for each facility in the 2020 National Emissions Inventory (NEI) (Personal communication with George Pouliot at the U.S. EPA). Using the average measured air temperature as the ambient temperature and the other CEMS data (United States Environmental Protection Agency (U.S. EPA), 2022), including hourly exit airflow rates, the morning and afternoon heat emissions are estimated as 52–59 and 49–56 MW, 80–92 and 76–87 MW, and 13–13 and 12–13 MW for Roxboro, Belews Creek, and CPI Roxboro, respectively. Note that the heat emission estimation is sensitive to the stack exit temperature, which is expected to vary from hour to hour, similar to the exit airflow rates and the $SO_2$ emissions. Nonetheless, these estimated values indicate the reasonable ranges of the heat emissions. When correlation-based and RMSE-based methods agree with each other in their optimal heat emission for Roxboro and Belews Creek morning segments, the optimal heat emissions are very close to the estimated stack heat emissions here. When the two methods disagree, the correlation-based optimal heat emissions of 90 and 140 MW for CPI Roxboro in the morning and afternoon are unreasonably high, but the RMSE-based optimal emissions of 30 and 50 MW could still be reasonable. This suggests that the RMSE-based results are probably more reliable.

While the uncertainty of the heat emission is the focus here, there are a lot of other uncertainties associated with the emission estimates. For instance, uncertainties in many parameters, such as the assumed background $SO_2$ mixing ratios, the meteorological data input such as the wind direction and speed, and some of the HYSPLIT turbulence parameterizations related to the turbulent mixing, will all affect the final results. Even if the hourly exit temperatures were available, the plume rise calculated using the Briggs algorithm may still misplace the plume. It is likely that the optimal heat emissions chosen here have compensated for other errors in the model.

The relatively low resolution of heat emissions with an increment of 10 MW for the plume rise ensemble runs may result in significant errors for some cases. For instance, Fig. 11 shows large ranges of the emission estimates when using 10 MW above and below the correlation-based and RMSE-based optimal heat emissions for the CPI Roxboro afternoon segment. Since it is not easy to select the best-performing plume rise run based on the limited observations, it is probably better to use several ensemble members to quantify the uncertainties of the model simulation as well as the emission estimates. This is indicated in Fig. 11 but needs to be further explored in the future.

Negative correlation is found between predictions and observations for the Belews Creek plume captured by the afternoon flight due to the wind direction errors of the meteorological data. However, the RMSE-based $SO_2$ emission estimate is only 2 % above the CEMS value. More surprisingly, if the plume rise run with the highest absolute correlation coefficient is selected, the $SO_2$ estimate of 715.6 kg h$^{-1}$ is very close to the CEMS average emission rate of 794 kg h$^{-1}$. We speculate that the inverse modeling is not very sensitive to the plume misplacement because the cost function minimization would favor an unbiased population distribution even when misplacement by the model is present. However, special care is needed for such situations in which large RMSEs indicate model deficiencies.

It has to be noted that the current dispersion simulation directly places the pollutant release points with the calculated plume rises elevated above the stacks, while the actual plumes reach their apexes gradually. Thus the dispersion model is not able to accurately reproduce the exact plume shapes at locations close to the source. The afternoon flight around Belews Creek power plant is closer to the source than the other segments. This probably makes this case more difficult to simulate accurately than the other segments.

This study shows that RMSE is a better metric than the correlation coefficient in choosing the best ensemble member for the $SO_2$ emission inversion. While the RMSE-based optimal plume rise runs appear to agree better with the observations than the correlation-based optimal runs, observations are often missing when and where the optimal runs are significantly different. Additional measurements at multiple altitudes would have been really helpful. In future flight planning for similar top-down emission estimation studies more vertical profiles of the target pollutant should be measured. In addition, more upwind measurements are also recommended in order to better quantify the background concentrations caused by many other emission sources. It is also wise to choose relatively steady meteorological conditions for the flight campaign since unsteady conditions such as frequent wind direction changes pose great challenges not only for the inverse modeling but also for the meteorological simulation and the dispersion modeling. The current study shows the value of ensemble simulations when certain model parameters are difficult to determine, such as stack heat emissions as shown here.

## Appendix A

**Table A1.** Estimated $SO_2$ emissions from the three power plants on 26 March 2019 with 15 different assumed heat emissions and the average CEMS emissions during the specified hours. A constant 0.199 ppbv background $SO_2$ mixing ratio is assumed and has been subtracted from the observations for emission inversion. The ranges of CEMS hourly emissions for the specified hours as well as 1 h before and 1 h after the period are shown after the average CEMS emission. The relevant CEMS hourly emissions are listed in Table 2. The bold numbers are associated with the heat emissions which generate the highest correlation coefficients between observations and HYSPLIT predictions for the specific flight segments. The italic number is associated with the heat emission that generates the highest absolute correlation coefficient between observations and HYSPLIT predictions for the Belews Creek afternoon segment.

| CEMS/assumed | Roxboro | | Belews Creek | | CPI Roxboro | |
|---|---|---|---|---|---|---|
| heat emission (MW) | 15:00–17:00Z (kg h$^{-1}$) | 19:00–20:00Z (kg h$^{-1}$) | 16:00–17:00Z (kg h$^{-1}$) | 18:00–19:00Z (kg h$^{-1}$) | 15:00–16:00Z (kg h$^{-1}$) | 19:00–20:00Z (kg h$^{-1}$) |
| CEMS | 429 (345–582) | 476 (465–509) | 905 (816–1132) | 794 (767–867) | 291 (279–306) | 306 (293–316) |
| 10 | 584.9 | 352.1 | 1556.8 | 1249.6 | 293.6 | 478.0 |
| 20 | 547.5 | 412.3 | 1397.6 | 922.3 | 288.9 | 491.5 |
| 30 | 519.8 | 416.4 | 1383.7 | 668.6 | 262.0 | 478.9 |
| 40 | 637.3 | 542.1 | 3333.0 | 452.0 | 471.3 | 415.6 |
| 50 | 513.6 | 382.9 | 1361.7 | 446.6 | 311.7 | 386.0 |
| 60 | 494.5 | 389.9 | 1300.6 | 335.6 | 317.3 | 370.9 |
| 70 | **436.0** | 413.1 | 1274.9 | 362.1 | 350.7 | 392.2 |
| 80 | 423.1 | 386.8 | **1259.7** | 556.4 | 388.3 | 348.6 |
| 90 | 427.4 | 376.8 | 1282.5 | 460.2 | **549.7** | 334.3 |
| 100 | 419.8 | **403.3** | 1247.1 | 590.7 | 886.2 | 350.7 |
| 110 | 416.4 | 410.6 | 1375.4 | *973.5* | 1151.4 | 323.1 |
| 120 | 421.5 | 455.6 | 1237.2 | 1808.5 | 1958.9 | 283.2 |
| 130 | 449.3 | 478.0 | 1504.8 | 4561.0 | 2606.8 | 282.1 |
| 140 | 409.0 | 614.8 | 1620.3 | 888.0 | 5682.1 | **294.7** |
| 150 | 417.3 | 708.5 | 1772.7 | 916.8 | 3267.0 | 311.1 |

**Table A2.** Estimated $SO_2$ emissions from the three power plants on 26 March 2019 with 15 different assumed heat emissions and the average CEMS emissions during the specified hours. The segment-specific minimum observations are assumed as the background $SO_2$ mixing ratios and have been subtracted from the observations for emission inversion. The ranges of CEMS hourly emissions for the specified hours as well as 1 h before and 1 h after the period are shown after the average CEMS emission. The relevant CEMS hourly emissions are listed in Table 2. The bold numbers are associated with the heat emissions which generate the highest correlation coefficients between observations and HYSPLIT predictions for the specific flight segments. The italic number is associated with the heat emission that generates the highest absolute correlation coefficient between observations and HYSPLIT predictions for the Belews Creek afternoon segment.

| CEMS/assumed | Roxboro | | Belews Creek | | CPI Roxboro | |
|---|---|---|---|---|---|---|
| heat emission (MW) | 15:00–17:00Z (kg h$^{-1}$) | 19:00–20:00Z (kg h$^{-1}$) | 16:00–17:00Z (kg h$^{-1}$) | 18:00–19:00Z (kg h$^{-1}$) | 15:00–16:00Z (kg h$^{-1}$) | 19:00–20:00Z (kg h$^{-1}$) |
| CEMS | 429 (345–582) | 476 (465–509) | 905 (816–1132) | 794 (767–867) | 291 (279–306) | 306 (293–316) |
| 10 | 700.1 | 448.4 | 1626.8 | 721.4 | 330.3 | 569.9 |
| 20 | 664.6 | 530.3 | 1463.3 | 537.8 | 330.3 | 576.7 |
| 30 | 634.8 | 527.2 | 1443.0 | 443.0 | 308.6 | 557.5 |
| 40 | 802.0 | 735.9 | 3503.7 | 350.7 | 541.0 | 488.6 |
| 50 | 617.3 | 488.6 | 1428.6 | 350.7 | 380.6 | 463.9 |
| 60 | 609.9 | 503.5 | 1361.7 | 291.2 | 383.7 | 443.9 |
| 70 | **550.8** | 527.2 | 1337.5 | 307.4 | 443.0 | 462.9 |
| 80 | 536.7 | 485.7 | **1316.3** | 407.4 | 481.8 | 416.4 |
| 90 | 518.8 | 482.8 | 1345.5 | 380.6 | **684.9** | 400.9 |
| 100 | 511.6 | **517.8** | 1308.4 | 499.5 | 1058.7 | 406.6 |
| 110 | 523.0 | 514.7 | 1443.0 | *725.7* | 1342.8 | 403.3 |
| 120 | 507.5 | 562.0 | 1300.6 | 1086.5 | 2147.5 | 352.1 |
| 130 | 517.8 | 590.7 | 1585.1 | 758.3 | 2575.7 | 347.9 |
| 140 | 468.5 | 784.5 | 1699.9 | 843.0 | 5214.3 | **371.6** |
| 150 | 468.5 | 873.9 | 1856.1 | 952.3 | 4127.4 | 387.5 |

**Table A3.** Estimated $SO_2$ emissions from the three power plants on 26 March 2019 with 15 different assumed heat emissions and the average CEMS emissions during the specified hours. The segment-specific 5th percentile observations are assumed as the background $SO_2$ mixing ratios and have been subtracted from the observations for emission inversion. The ranges of CEMS hourly emissions for the specified hours as well as 1 h before and 1 h after the period are shown after the average CEMS emission. The relevant CEMS hourly emissions are listed in Table 2. The bold numbers are associated with the heat emission which generates the highest correlation coefficients between observations and HYSPLIT predictions for the specific flight segments. The italic number is associated with the heat emissions that generate the highest absolute correlation coefficient between observations and HYSPLIT predictions for the Belews Creek afternoon segment.

| CEMS/assumed | Roxboro | | Belews Creek | | CPI Roxboro | |
|---|---|---|---|---|---|---|
| heat emission (MW) | 15:00–17:00Z $(kg\,h^{-1})$ | 19:00–20:00Z $(kg\,h^{-1})$ | 16:00–17:00Z $(kg\,h^{-1})$ | 18:00–19:00Z $(kg\,h^{-1})$ | 15:00–16:00Z $(kg\,h^{-1})$ | 19:00–20:00Z $(kg\,h^{-1})$ |
| CEMS | 429 (345–582) | 476 (465–509) | 905 (816–1132) | 794 (767–867) | 291 (279–306) | 306 (293–316) |
| 10 | 668.6 | 433.4 | 1489.9 | 701.5 | 320.5 | 550.8 |
| 20 | 638.6 | 511.6 | 1334.8 | 523.0 | 318.6 | 560.8 |
| 30 | 603.9 | 509.5 | 1329.5 | 430.8 | 298.3 | 542.1 |
| 40 | 752.3 | 701.5 | 3326.3 | 342.4 | 523.0 | 473.2 |
| 50 | 589.6 | 470.4 | 1300.6 | 342.4 | 365.0 | 447.5 |
| 60 | 581.4 | 485.7 | 1244.7 | 284.3 | 368.7 | 430.0 |
| 70 | **518.8** | 511.6 | 1217.6 | 300.1 | 424.8 | 449.3 |
| 80 | 506.5 | 470.4 | **1210.3** | 395.4 | 462.0 | 402.5 |
| 90 | 487.6 | 466.7 | 1227.4 | 371.6 | **659.4** | 388.3 |
| 100 | 485.7 | **502.5** | 1191.1 | 490.6 | 1017.2 | 393.8 |
| 110 | 494.5 | 498.5 | 1316.3 | *711.4* | 1292.8 | 390.6 |
| 120 | 481.8 | 545.4 | 1179.3 | 1058.7 | 2092.4 | 341.7 |
| 130 | 495.5 | 573.3 | 1437.2 | 731.5 | 2474.8 | 337.6 |
| 140 | 446.6 | 756.8 | 1566.2 | 828.0 | 5090.8 | **359.9** |
| 150 | 448.4 | 849.8 | 1723.8 | 937.2 | 3895.0 | 375.3 |

**Table A4.** Estimated $SO_2$ emissions from the three power plants on 26 March 2019 with 15 different assumed heat emissions and the average CEMS emissions during the specified hours. The segment-specific 10th percentile observations are assumed as the background $SO_2$ mixing ratios and have been subtracted from the observations for emission inversion. The ranges of CEMS hourly emissions for the specified hours as well as 1 h before and 1 h after the period are shown after the average CEMS emission. The relevant CEMS hourly emissions are listed in Table 2. The bold numbers are associated with the heat emission which generates the highest correlation coefficients between observations and HYSPLIT predictions for the specific flight segments. The italic number is associated with the heat emission that generates the highest absolute correlation coefficient between observations and HYSPLIT predictions for the Belews Creek afternoon segment.

| CEMS/assumed | Roxboro | | Belews Creek | | CPI Roxboro | |
|---|---|---|---|---|---|---|
| heat emission (MW) | 15:00–17:00Z $(kg\,h^{-1})$ | 19:00–20:00Z $(kg\,h^{-1})$ | 16:00–17:00Z $(kg\,h^{-1})$ | 18:00–19:00Z $(kg\,h^{-1})$ | 15:00–16:00Z $(kg\,h^{-1})$ | 19:00–20:00Z $(kg\,h^{-1})$ |
| CEMS | 429 (345–582) | 476 (465–509) | 905 (816–1132) | 794 (767–867) | 291 (279–306) | 306 (293–316) |
| 10 | 655.4 | 415.6 | 1329.5 | 708.5 | 309.8 | 514.7 |
| 20 | 623.5 | 490.6 | 1186.4 | 525.0 | 306.8 | 526.1 |
| 30 | 589.6 | 487.6 | 1191.1 | 431.7 | 284.9 | 507.5 |
| 40 | 733.0 | 666.0 | 3107.9 | 342.4 | 488.6 | 442.2 |
| 50 | 572.2 | 452.0 | 1146.8 | 341.7 | 344.4 | 417.3 |
| 60 | 565.3 | 463.9 | 1088.7 | 283.2 | 352.1 | 400.9 |
| 70 | **503.5** | 489.6 | 1065.0 | 298.9 | 401.7 | 422.3 |
| 80 | 488.6 | 452.0 | **1067.2** | 393.8 | 436.9 | 376.1 |
| 90 | 472.3 | 446.6 | 1071.4 | 371.6 | **628.5** | 364.3 |
| 100 | 472.3 | **481.8** | 1033.6 | 490.6 | 969.6 | 371.6 |
| 110 | 478.9 | 479.9 | 1126.3 | *715.6* | 1242.2 | 365.0 |
| 120 | 468.5 | 525.0 | 1021.3 | 1069.3 | 2030.7 | 321.2 |
| 130 | 485.7 | 551.9 | 1247.1 | 741.8 | 2411.4 | 316.1 |
| 140 | 435.1 | 725.7 | 1367.2 | 828.0 | 5020.1 | **335.6** |
| 150 | 436.9 | 821.4 | 1519.9 | 939.1 | 3624.7 | 352.8 |

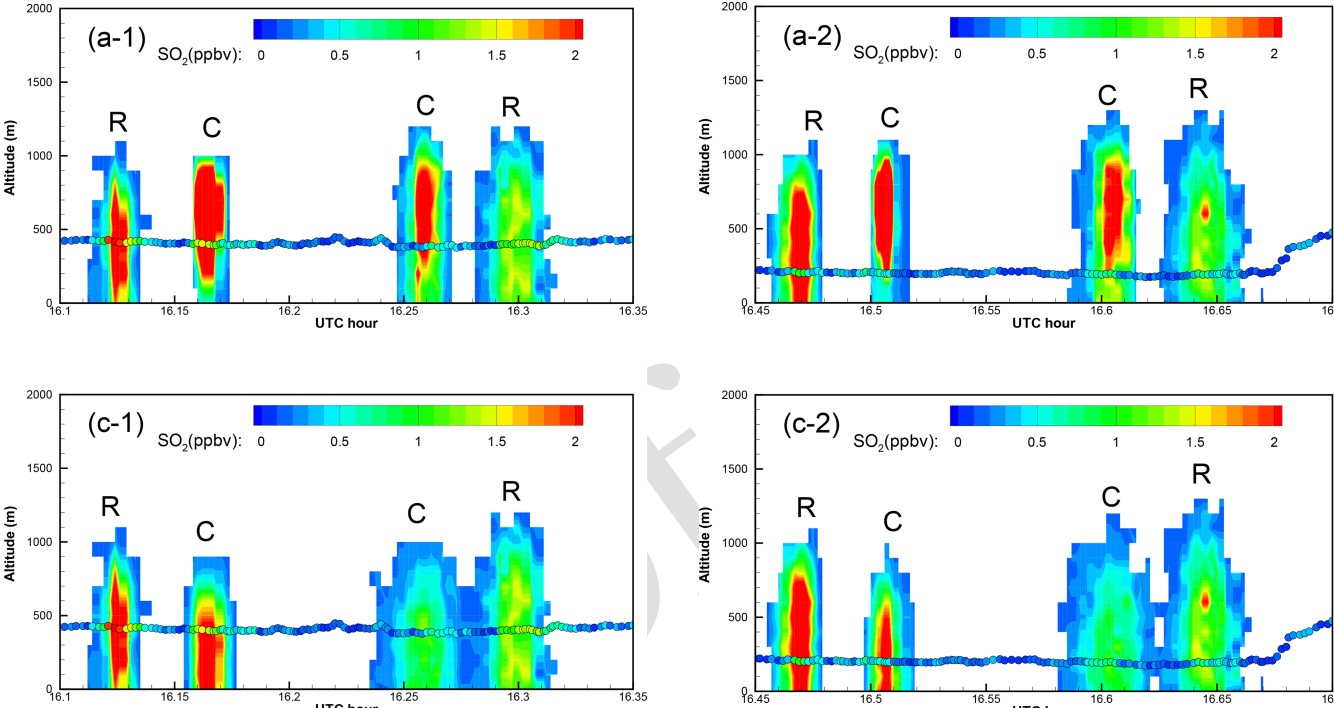

**Figure A1.** Enlarged "curtain" plots of the correlation-based **(a)** and the RMSE-based **(c)** "optimal" predictions in Fig. 9 (Part 1). Five portions of the morning curtain plot are enlarged, with the earliest period as Part 1 and the latest one as Part 5. In the curtain plots, continuous vertical profiles along the flight track are shown following the observation time. The morning flight 1 min observations are overlaid as circles. Color indicates the $SO_2$ values for both predictions and observations. Predicted plumes from Roxboro, Belews Creek, and CPI Roxboro are indicated with letters R, B, and C, respectively.

**Figure A2.** Enlarged "curtain" plots of the correlation-based **(a)** and the RMSE-based **(c)** "optimal" predictions in Fig. 9 (Part 2). Five portions of the morning curtain plot are enlarged, with the earliest period as Part 1 and the latest one as Part 5. In the curtain plots, continuous vertical profiles along the flight track are shown following the observation time. The morning flight 1 min observations are overlaid as circles. Color indicates the $SO_2$ values for both predictions and observations. Predicted plumes from Roxboro, Belews Creek, and CPI Roxboro are indicated with letters R, B, and C, respectively.

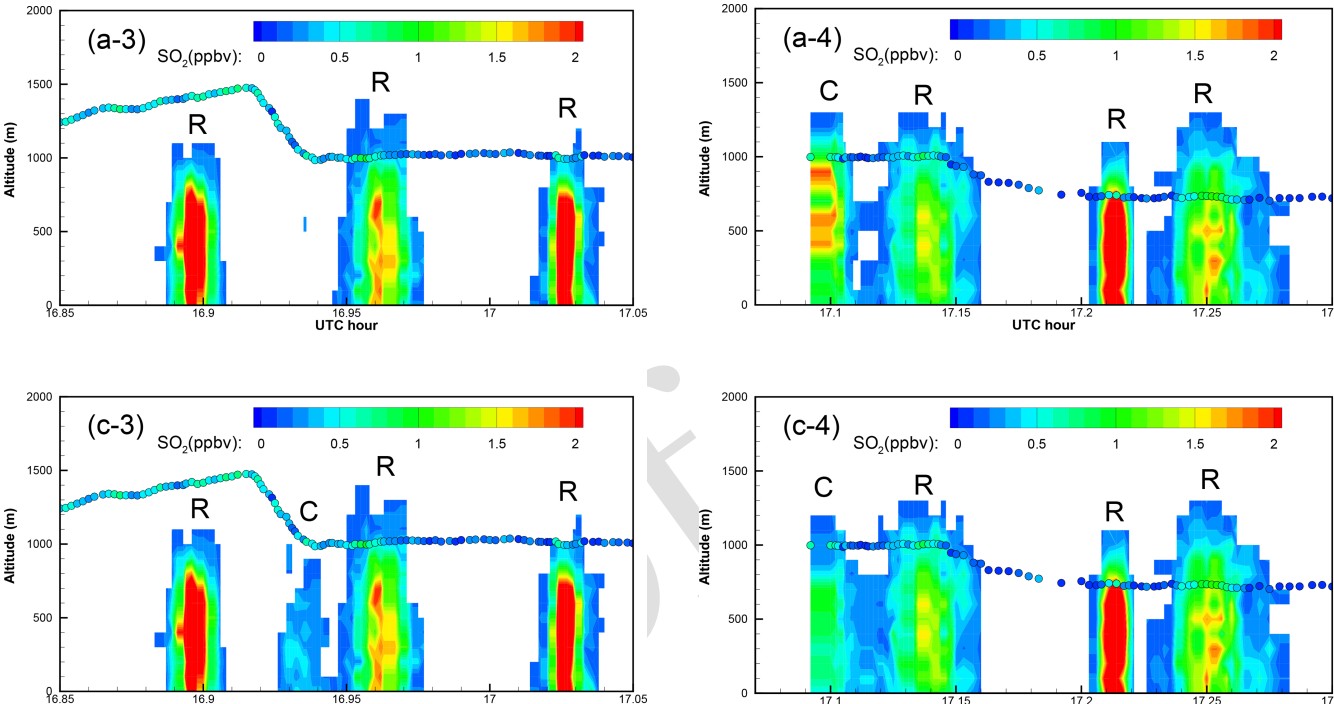

**Figure A3.** Enlarged "curtain" plots of the correlation-based **(a)** and the RMSE-based **(c)** "optimal" predictions in Fig. 9 (Part 3). Five portions of the morning curtain plot are enlarged, with the earliest period as Part 1 and the latest one as Part 5. In the curtain plots, continuous vertical profiles along the flight track are shown following the observation time. The morning flight 1 min observations are overlaid as circles. Color indicates the $SO_2$ values for both predictions and observations. Predicted plumes from Roxboro, Belews Creek, and CPI Roxboro are indicated with letters R, B, and C, respectively.

**Figure A4.** Enlarged "curtain" plots of the correlation-based **(a)** and the RMSE-based **(c)** "optimal" predictions in Fig. 9 (Part 4). Five portions of the morning curtain plot are enlarged, with the earliest period as Part 1 and the latest one as Part 5. In the curtain plots, continuous vertical profiles along the flight track are shown following the observation time. The morning flight 1 min observations are overlaid as circles. Color indicates the $SO_2$ values for both predictions and observations. Predicted plumes from Roxboro, Belews Creek, and CPI Roxboro are indicated with letters R, B, and C, respectively.

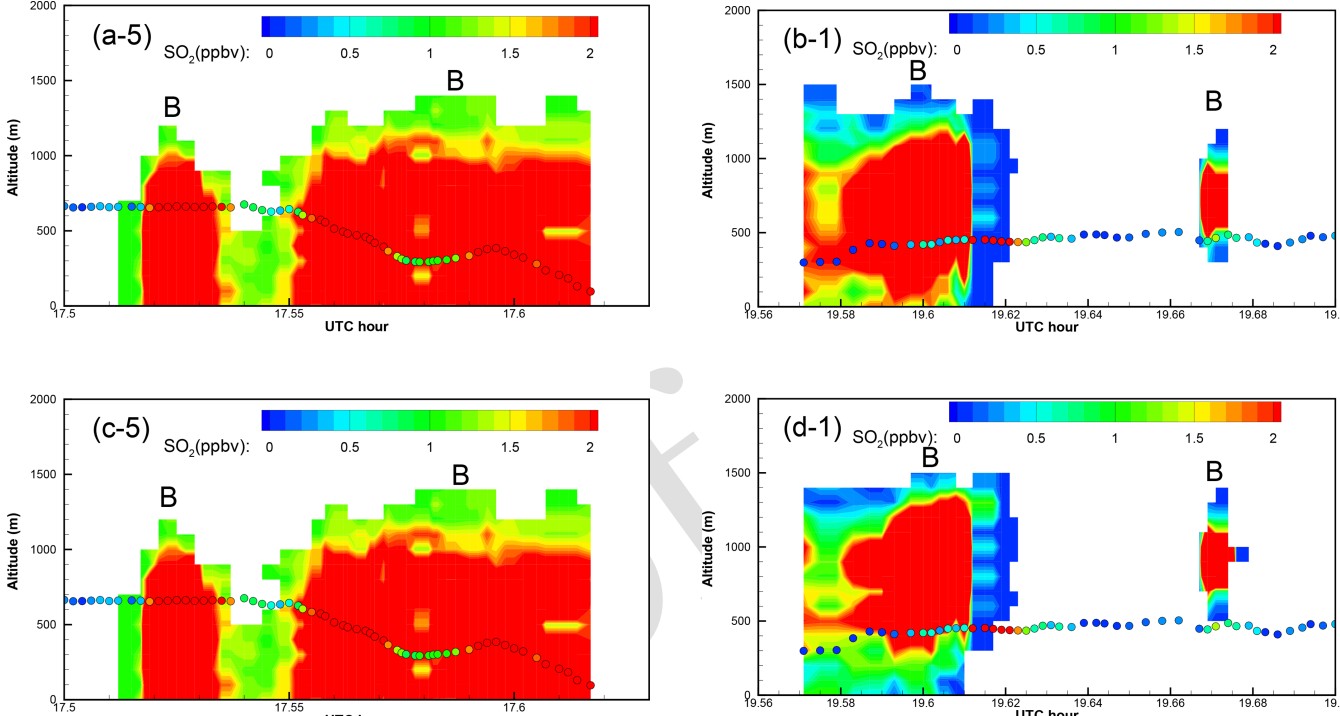

**Figure A5.** Enlarged "curtain" plots of the correlation-based **(a)** and the RMSE-based **(c)** "optimal" predictions in Fig. 9 (Part 5). Five portions of the morning curtain plot are enlarged, with the earliest period as Part 1 and the latest one as Part 5. In the curtain plots, continuous vertical profiles along the flight track are shown following the observation time. The morning flight 1 min observations are overlaid as circles. Color indicates the $SO_2$ values for both predictions and observations. Predicted plumes from Roxboro, Belews Creek, and CPI Roxboro are indicated with letters R, B, and C, respectively.

**Figure A6.** Enlarged "curtain" plots of the correlation-based **(b)** and the RMSE-based **(d)** "optimal" predictions in Fig. 9 (Part 1). Five portions of the morning curtain plot are enlarged, with the earliest period as Part 1 and the latest one as Part 5. In the curtain plots, continuous vertical profiles along the flight track are shown following the observation time. The afternoon flight 1 min observations are overlaid as circles. Color indicates the $SO_2$ values for both predictions and observations. Predicted plumes from Roxboro, Belews Creek, and CPI Roxboro are indicated with letters R, B, and C, respectively.

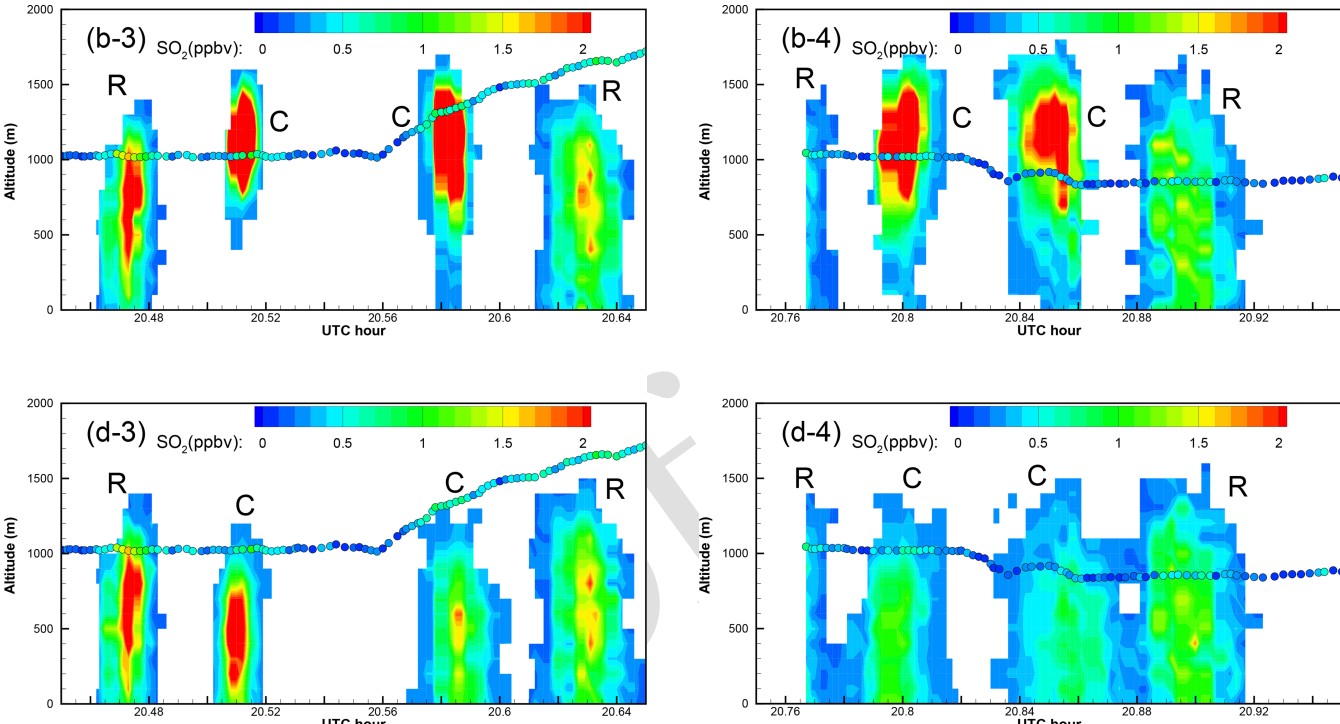

**Figure A7.** Enlarged "curtain" plots of the correlation-based **(b)** and the RMSE-based **(d)** "optimal" predictions in Fig. 9 (Part 3). Five portions of the morning curtain plot are enlarged, with the earliest period as Part 1 and the latest one as Part 5. In the curtain plots, continuous vertical profiles along the flight track are shown following the observation time. The afternoon flight 1 min observations are overlaid as circles. Color indicates the $SO_2$ values for both predictions and observations. Predicted plumes from Roxboro, Belews Creek, and CPI Roxboro are indicated with letters R, B, and C, respectively.

**Figure A8.** Enlarged "curtain" plots of the correlation-based **(b)** and the RMSE-based **(d)** "optimal" predictions in Fig. 9 (Part 4). Five portions of the morning curtain plot are enlarged, with the earliest period as Part 1 and the latest one as Part 5. In the curtain plots, continuous vertical profiles along the flight track are shown following the observation time. The afternoon flight 1 min observations are overlaid as circles. Color indicates the $SO_2$ values for both predictions and observations. Predicted plumes from Roxboro, Belews Creek, and CPI Roxboro are indicated with letters R, B, and C, respectively.

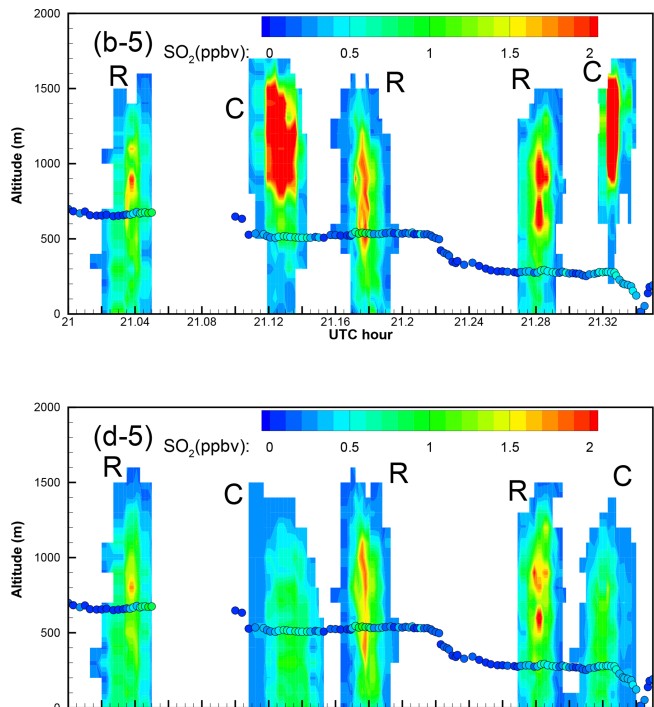

**Figure A9.** Enlarged "curtain" plots of the correlation-based **(b)** and the RMSE-based **(d)** "optimal" predictions in Fig. 9 (Part 5). Five portions of the morning curtain plot are enlarged, with the earliest period as Part 1 and the latest one as Part 5. In the curtain plots, continuous vertical profiles along the flight track are shown following the observation time. The afternoon flight 1 min observations are overlaid as circles. Color indicates the $SO_2$ values for both predictions and observations. Predicted plumes from Roxboro, Belews Creek, and CPI Roxboro are indicated with letters R, B, and C, respectively.

**Code and data availability.** HYSPLIT code is available at https://www.ready.noaa.gov/HYSPLIT.php (NOAA, 2023). The observation data are available upon request.

**Author contributions.** TC designed and performed the model analysis and wrote the first draft of the paper. XR conducted the measurement collection and analysis. FN completed the WRF runs to generate meteorological data. MC provided expertise for the HYSPLIT modeling and plume rise algorithm. AC conducted the initial $SO_2$ simulations. All authors contributed to the paper editing and revision.

**Competing interests.** The contact author has declared that none of the authors has any competing interests.

**Disclaimer.** Publisher's note: Copernicus Publications remains neutral with regard to jurisdictional claims in published maps and institutional affiliations.

**Acknowledgements.** This study was supported by NOAA award NA16OAR4590121 at the NOAA Air Resources Laboratory in collaboration with the Cooperative Institute for Satellites Earth System Studies (CISESS), University of Maryland, College Park, MD 20740, USA.

**Financial support.** This research has been supported by NOAA Research (NOAA award no. NA16OAR4590121).

**Review statement.** This paper was edited by Joshua Fu and reviewed by two anonymous referees.

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

## Remarks from the typesetter