# Peer review of "Estimation of power plant SO2 emissions using HYSPLIT dispersion model and airborne observations with plume rise ensemble runs"

_EGUsphere, 2023_

## Author Comment (AC1)

**Estimation of power plant $SO_2$ emissions using HYSPLIT dispersion model and airborne observations with plume rise ensemble runs**

Tianfeng Chai, Xinrong Ren, Fong Ngan, Mark Cohen, and Alice Crawford

**Response to the comments of Reviewer 1**

**July 31, 2023**

The paper describes attempts to estimate SO2 emissions from power plants by use of a Lagrangian dispersion model and aircraft measurements. It emphasizes the uncertainty in plume rise due to stack heat input, which is treated as unknown. Two methods are used to find the optimal heat input. There is some good information here, but the presentation could be clearer, and the implications should be more clearly stated.

We thank the referee for thoroughly reading the manuscript and providing valuable comments. The manuscript has been revised for a better presentation of the objective, findings, and the implications.

Point-by-point responses to the referee's specific comments are given below.

**General comments:**

1) The objective of the paper seems to be to find ways to determine the optimum simulation to produce the correct (known) emissions. Two methods are suggested, one based on correlation between the observed and simulated time series, and the other based on the RMS difference of that same time series. Unfortunately I have just explained the objective more clearly than the paper ever does. These are reasonable proposals for how to determine the optimum simulation, but they both have flaws, which are evident in the data. For example, both the correlation and the RMS are sensitive to misplacement of the plume, whereas the inversion may not be sensitive to that misplacement.

Thanks for pointing this shortcoming in the original manuscript. We have revised the paper to make the presentation clear. In particular, the abstract has been rewritten to better explain the objective of the study. Some of the details from the abstract has been removed to emphasize the main points of the paper as explained by the reviewer here.

2) The heat input to the plume rise calculation is treated as a free parameter. There must be reasonable estimates of the real value available, based on the CEMS data and the characteristics of the plants, for example whether they have scrubbers or not. If the optimization process finds values that are well outside a reasonable range, that may indicate that the plume rise calculation is inadequate, which would be valuable information.

The following paragraph has been added to the Summary and discussion section.

While the stack exit gas temperature data are not available for this study, a single constant stack exit temperature is provided for each facility in the 2020 National Emissions Inventory (NEI) (Personal communication with George Pouliot at the U.S. EPA). Using the average measured air temperature as the ambient temperature and the other CEMS data (United States Envi- ronmental Protection Agency (U.S. EPA), 2022), including hourly exit air flow rates, the morning/afternoon heat emissions are estimated as 52–59 MW/49–56 MW, 80–92 MW/76–87 MW, and 13–13 MW/12–13 MW, for Roxboro, Belews Creek, and CPI Roxboro, respectively. Note that the heat emission estimation is sensitive to the stack exit temperature which is expected to vary from hour to hour, similar to the exit air flow rates and the  $SO_2$  emissions. Nonetheless, these estimated values indicate the reasonable ranges of the heat emission. When correlation-based and RMSE-based methods agree with each other in their "optimal" heat emission for Roxboro and Belews Creek morning segments, the "optimal" heat emissions are very close to the estimated stack heat emissions here. When the two methods disagree, the correlation-based "optimal" heat emissions of 90 MW/140 MWfor CPI Roxboro morning/afternoon are unreasonably high, but the RMSE-based "optimal" emissions of 30 MW/50 MW could still be reasonable. This suggests that the RMSE-based results are probably more reliable.

3) Only two of the many possible sources of uncertainty are explored here. That's fine if it is clearly stated. The two sources examined are the plume rise and the background specification. Errors in wind direction are present, and get some attention. Errors in vertical placement and mixing of the plume are probably also present, but are not discussed at all. As it stands, varying the heat input amounts to looking for the value that best compensates other errors in the model. That's not wrong as an empirical method, but again, it should be stated.

In the previous version, the uncertainty issues pointed out here were mentioned in the Summary and discussion section, but were not very clear. The statements have been modified and moved into a separate paragraph, as shown below. While the uncertainty of the heat emission is focused here, there are a lot of other uncertainties associated with the emission estimates. For instance, uncertainties in many parameters, such as the assumed background  $SO_2$  mixing ratios, the meteorological data input such as the wind direction and speed, and some of the HYSPLIT turbulence parameterizations related to the turbulent mixing, will all affect the final results. Even if the hourly exit temperature were available, the plume rise calculated using the Briggs algorithm may still misplace the plume. It is likely that the "optimal" heat emissions chosen here have compensated other errors in the model.

4) The vertical structure of the simulated plumes should get more emphasis. Some of the figures in the Appendix should be promoted to the main text. It looks like the flights were rather close to the plants, that is, in the region where the plume is not well-mixed in the vertical. This is arguably a mistake in the flight planning, unless it is an error in the model (too slow mixing). In theory, the inversion should recover the correct emissions as long as the observation samples a reasonable amount of the plume, but this is a very strong constraint on the precision of the simulation.

We have moved the previous Figure A7 to the main text (as Figure 10 in the revision). While it is presumably true that the inversion should be able to estimate the emissions as long as the observation samples a reasonable amount of the plume, it is difficult for the inverse modeling to accurately estimate the emissions due to many other uncertainties in the model simulation, as pointed out by the reviewer. The statement, "The results here indicate the need to have more observations at different altitudes in the future flight planning", has been added after presenting all the results.

5) It seems that we are to take the set of differences between the inverted and known emissions as a measure of the uncertainty of the method, but this is never stated. Although a formal uncertainty analysis is not really possible with such a small number of samples, some statement should be made. Clearly the differences are not Gaussian, and the large differences (which may or may not be "outliers"), are of concern.

We have added the following statements when presenting the results in Figure 10 (now Figure 11 in the revision).

Note that the ranges of the inverted emissions with 10 MW above and below the "optimal" heat emissions are used as to indicate the sensitivities of the results to the heat emissions. While the differences between the emission estimates and the known CEMS data provide some confidence to the results, quantification of the uncertainties associated with the method probably requires further investigation in the future.

6) More detail is needed on the WRF runs. WRF has many options. The chosen options, initial and boundary data, etc. must be stated in enough detail that WRF experts can judge whether they are reasonable, and others can plausibly replicate the results. In particular, whether using the mixed layer depth (PBLH?) out of WRF directly is reasonable depends on the physics options chosen.

The following has been added to address the missing detail.

The WRF model was configured for three-nested domains with horizontal grid spacing of 27 km (D01), 9 km (D02), and 3 km (Figure 2). A total of 33 vertical layers were defined with a higher resolution near the surface and 100 hPa for model 110 top. There were 20 layers below 850 hPa with the first mid-layer height of the model at around 8 m. The simulations for the D01 were initialized by using the North American Regional Reanalysis (Mesinger et al., 2006) with 32-km grid spacing and available every 3 h. Then, the WRF results from the coarser domains provided the initial and boundary conditions for the inner domains. The daily WRF runs had a 30-hr duration including 6-hr a spin-up period (i.e., starting at 18 UTC on the previous day). The physics options for the WRF simulations were - the rapid radiative transfer model for radiation parameterization 115 (Iacono et al., 2008), WSM6 for microphysics i(Lim and Hong, 2010), the Grell 3D Ensemble for the sub-grid cloud scheme (Grell and Devenyi, 2002), Noah land-surface model (Chen and Dudhia, 2001), and Mellor-Yamada-Nakanishi-Niino 2.5 level TKE scheme for the planetary boundary layer (PBL) parameterization and its corresponding surface layer scheme (Nakanishi and Niino, 2006).

Mesinger, F., DiMego, G., Kalnay, E., Mitchell, K., Shafran, P., Ebisuzaki, W., Jovic, D., Woollen, J., Rogers, E., Berbery, E., Ek, M., Fan, Y., Grumbine, R., Higgins, W., Li, H., Lin, Y., Manikin, G., Parrish, D., and Shi, W.: North American regional reanalysis, Bull. Amer. Meteorol. Soc., 87, 343–360, https://doi.org/10.1175/BAMS-87-3-343, 2006.

Iacono, M. J., Delamere, J. S., Mlawer, E. J., Shephard, M. W., Clough, S. A., and Collins, W. D.: Radiative forcing by long-lived greenhouse gases: Calculations with the AER radiative transfer models, J. of Geophys. Res., 113, https://doi.org/10.1029/2008JD009944, 2008.

Lim, K.-S. S. and Hong, S.-Y.: Development of an Effective Double-Moment Cloud Microphysics Scheme with Prog- nostic Cloud Condensation Nuclei (CCN) for Weather and Climate Models, Monthly Weather Review, 138, 1587–1612, https://doi.org/10.1175/2009MWR2 2010. Chen, F. and Dudhia, J.: Coupling an advanced land surface-hydrology model with the Penn State-NCAR MM5 modeling system. Part I: Model implementation and sensitivity, Monthly Weather Review, 129, 569–585, 2001.

Grell, G. and Devenyi, D.: A generalized approach to parameterizing convection combining ensemble and data assimilation techniques, Geophys. Res. Lett., 29, https://doi.org/10.1029/2002GL015311, 2002.

Nakanishi, M. and Niino, H.: An improved Mellor-Yamada level-3 model: Its numerical stability and application to a regional prediction of advection fog, Bound.-Layer Meteorol., 119, 397–407, https://doi.org/10.1007/s10546-005-9030-8, 2006.

7) The SO2 background is clearly important in this region. I recommend simplifying the presentation by removing the parts where background is not used. Furthermore, I am concerned that the method used to derive the background may be chosen primarily because it gives the best results (given compensating errors). The explanation is not perfectly clear, but the 25th percentile within the plume seems like it should yield a value considerably higher than background, which is usually taken to be outside the plume.

We like to keep the results where background is not used in order to emphasize the importance of the including the background in the analyses. Since the choice of the background value is not trivial, we presented several options to estimate the background. The choice to take the segment-specific minimum as the background is actually very close to take the measurement outside the plume. In this study, the 25th percentile choice yields the best inversion results as shown in Table 5. It is possible that is partially due to some compensating errors. We do not imply that the 25th percentile choice would be universally suitable to other cases.

8) The conclusions should state the authors' recommendations for future studies. This should include a recommendation for which optimization method to use, or that another method is needed. Guidelines for deciding whether a given set of observations is useful or should be discarded would be helpful. Does a large RMS relative to the mean imply that a flight should be discarded? Implications for flight planning should be included. Do the authors recommend using single deterministic meteorology, or should ensembles be used?

We have revised the conclusion. In particular, the following two paragraphs have been added. The first one gives more evidence to show that the RMSE-based results are more reliable. The last paragraph implemented the reviewer's recommendation on the needed content. The third from last paragraph in the revision (not copied here) partially addressed "the guidelines for Guidelines for deciding whether a given set of observations is useful or should be discarded" and the question of "a large RMS relative to the mean imply that a flight should be discarded". We have added the statement, "However, special care is needed for such situations where large RMSEs also indicate the model deficiencies", at the end of the paragraph.

While the stack exit gas temperature data are not available for this study, a single constant stack exit temperature is provided for each facility in the 2020 National Emissions Inventory (NEI) (Personal communication with George Pouliot at the U.S. EPA). Using the average measured air temperature as the ambient temperature and the other CEMS data (United States Environmental Protection Agency (U.S. EPA), 2022), including hourly exit air flow rates, the morning/afternoon heat emissions are estimated as 52–59 MW/49–56 MW, 80–92 MW/76–87 MW, and 13–13 MW/12–13 MW, for Roxboro, Belews Creek, and CPI Roxboro, respectively. Note that the heat emission estimation is sensitive to the stack exit temperature which is expected to vary from hour to hour, similar to the exit air flow rates, and the  $SO_2$  emissions. Nonetheless, these estimated values indicate the reasonable ranges of the heat emission. When correlation-based and RMSE-based methods agree with each other in their "optimal" heat emission for Roxboro and Belews Creek morning segments, the "optimal" heat emissions are very close to the estimated stack heat emissions here. When the two methods disagree, the correlation-based "optimal" heat emissions of 90 MW/140 MW for CPI Roxboro morning/afternoon are unreasonably high, but the RMSE-based "optimal" emissions of 30 MW/50 MW could still be reasonable. This suggests that the RMSE-based results are probably more reliable.

This study shows that RMSE is a better metric than correlation coefficient in choosing the best ensemble member for the  $SO_2$  emission inversion. While the RMSEbased "optimal" plume rise runs appear to agree better with the observations than the correlation-based "optimal" runs, observations are often missing when and where the "optimal" runs are significantly different. Additional measurements at mulitiple altitudes would have been really helpful. In the future flight planning of similar topdown emission estimation studies more vertical profiles of the target pollutant should be measured. In addition, more upwind measurements are also recommended in order to better quantify the background concentrations caused by many other emission sources. It is also wise to choose relative steady meteorological conditions for the flight campaign since the unsteady conditions such as frequent wind direction changes pose great challenges not only to the inverse modeling but also to the meteorological simulation and the dispersion modeling. The current study shows the value of the ensemble simulations when certain model parameters are difficult to determine, such as stack heat emission here.

**Specific comments:**

1) The abstract is long and detailed, but does not clearly state the objectives or method. It is more of an introduction than an abstract.

The abstract has been modified. The objective of the study is now explicitly stated at the beginning of the abstract. Some details are also removed to better reflect on the findings of the study.

2) Line 273: The standard deviation of the 1-s observations is not a reasonable estimate of the observational uncertainty. It does not take into account sampling error (probably dominant here). How is the process affected by a larger observation uncertainty estimate?

In our previous study (Chai et al. 2018), it was found that the emission estimates were not very sensitive to the observation uncertainty estimates (Table 3 in Chai et al. 2018). This has been clarified in the revision.

Chai, T., Stein, A., and Ngan, F.: Weak-constraint inverse modeling using HYSPLIT-4 Lagrangian dispersion model and Cross-Appalachian Tracer Experiment (CAPTEX) observations – effect of including model uncertainties on source term estimation, Geosci. Model Dev., 11, 5135–5148, https://doi.org/10.5194/gmd-11-5135-2018, 2018

---

## Author Comment (AC2)

**Estimation of power plant $SO_2$ emissions using HYSPLIT dispersion model and airborne observations with plume rise ensemble runs**

Tianfeng Chai, Xinrong Ren, Fong Ngan, Mark Cohen, and Alice Crawford

**Response to the comments of Reviewer 2**

July 31, 2023

*My overall rating is accept only after major revisions. I think that the use of the retrieval algorithm has merit, and that's the key reason I'm not recommending rejection. However, there are several issues with the paper that need to be resolved before I can recommend acceptance. Examples (details follow):*

We thank the reviewer for reading the manuscript thoroughly and providing the insightful comments and suggestions. Point-to-point responses are provided after the examples and detailed comments.

- *Reason for the need for Qh estimates rather than stack emission temperatures and vertical velocities is unclear. The latter are usually part of CEMS observations, and emissions inventories usually include these parameters as time dependent values or annual averages. I've included suggested sources of information and an EPA emissions inventory staff person's email address to contact on information for the sources the authors studied.*

  We've contacted the EPA inventory staff, George Pouliot, as suggested by the reviewer. In our email exchanges, he stated that "the (hourly) stack exit temperature is not routinely reported to the EPA as part of the CEMS data." While a single constant stack exit temperature is given in the annual NEI, it is definitely not a constant number and can vary from hour to hour. During our investigation, we did contact all the three power plants for the hourly exit temperature data but could not get those information. Using the representative stack exit temperature from the 2020 NEI provided by George Pouliot and flight observations, we estimated the heat emissions of the six segments. The results are included and discussed in the revision.

- *Details on the plume-rise calculation need to be given – there are different ways of implementing Briggs algorithms (see references given below).*

Details on the plume rise calculation are provided in Equations (1)-(3). More details can be found in the following two references, Arya (1999) and Briggs (1969). The recommended reference of Briggs (1984) has been added.

Arya, 460 S. P.: Air pollution meteorology and dispersion, Oxford University Press, New York, NY, 1999.

Briggs, G. A.: Plume Rise, AEC Critical Review Series TID-25075, U.S. Atomic Energy Commission, Division of Technical Information, Oak Ridge, Tennessee, 1969

- *The extent to which the observations indicate conditions suitable for attempting retrievals (steady-state of the observation data) is unclear – but this could be determined from the observation data (references to consult are included).*

We believe that steady state is not required for the current inversion method since the dispersion model can represent the temporal variations. However, when the variations are not well represented by the model, it brings in large uncertainties to the emission estimates. This has been commented in the paper at several places, e.g. in "Summary and discussion" section.

- *The extent to which the HYSPLIT model provides sufficient process detail for a reactive gas such as SO2 for determination of emissions estimates is unclear. There is a risk that this model is too simplified to adequately simulate SO2 concentrations, and the lack of process detail may contribute to emissions estimate errors.*

Green et al. (2019) found that the SO 2 oxidation rates during the day from power plants were 0.22–0.71%/hr, using 13 flights from 6 February to 15 March 2015 over the eastern United States. The measurements were made during a clear-sky day on March 26, 2019 and the travel time of the measured air parcels from the stacks are less than three hours. So it is reasonable to treat $SO_2$ as a passive tracer and ignore its oxidation. This has been clarified in the revision. In addition, "more upwind measurements are also recommended in order to better quantify the background concentrations caused by many other emission sources" has been explicitly stated for the future flight planning in "Summary and discussion" section.

- *The measurement data did not attempt to bracket the individual sources, which results in ambiguity regarding upwind concentrations and the possibility for meteorological conditions to be changing in the source region. The authors can not do much about this at this stage, but the issue should be acknowledged and the Introduction should include*

*a review of other aircraft studies flight path methodology and a discussion on how this may affect retrieval results.*

The findings by Fathi et al. (2021) on the impact of storage-and-release have been acknowledged in the introduction when discussing aircraft studies, with the following statements.

"Fathi et al. (2021) investigated the impact of storage-and-release due to meteorological variability on mass-balance emission rate retrieval accuracy using virtual aircraft sampling of a regional chemical transport model output. The storage-and-release events contributed to the mass-balance emission estimate errors ranging from -25% to 24% in their tests. They recommended repeat flights around the given facility and/or time-consecutive upwind and downwind vertical profiling during the sampling period."

While box or oval flights that bracket the individual sources would definitely help, we believe they are not critical to the current method where the relationship between sources and receptors are provided by the dispersion model. The inverse modeling method using a dispersion model without assuming constant meteorological fields is expected to perform better than the mass balance method. This has been clarified.

**Detailed comments:**

- *Abstract, pages 1 and 2, lines 1 to 30: Some rewording of the abstract is needed. The abstract provides a step-by-step description of "what was done" but does not describe the goal of the project. For example, was the objective of the work to test out the TCM retrieval method on aircraft observations (that is, was the objective to determine whether or not the method works and how well if so), was it to determine the circumstances under which retrievals can be carried out, etc.? The reader of the abstract needs to be told why the work was being done, whether or not the project was successful (why/why not), and the main conclusions resulting from the project. The authors have focused on the fine grain detail of the work and not on the big picture, which should be the focus.*

  The abstract has been rewritten following the suggestions here. The objective of the study is now explicitly stated at the beginning of the abstract. Some fine grain details are also removed to better reflect on the findings of the study.

- *Introduction, page 2, line 45, line 62: The authors' list of "source term estimation applications have been developed using various dispersion models and inverse modeling*

*schemes" misses a few recent ones appearing in Atmospheric Chemistry and Physics and other journals such as Nature. E.g. Fathi et al 2021 (https://acp.copernicus.org/articles/21/1546 describes meteorological conditions under which retrievals of SO2 emissions from aircraft observations are likely to result in significant errors in retrieved emissions, as well as some of the implementation details of dispersion models which may lead to errors in retrievals if they are not recognized and taken into account. That is, some meteorological conditions may result in erroneous emissions estimates – these may explain some of the authors' problems with some of their aircraft retrievals. One underlying concept for the successful retrievals explored in the above-referenced work is that the meteorology approximates a steady-state during the time of the retrieval – the direction and speed of the winds, the change in wind speed with height, and the atmospheric stability are all invariant with time as the observations are taking place. Do the authors have sufficient data to determine whether the meteorological conditions were stable during the observation time (and does variability in those conditions explain for example the negative correlation between retrievals and CEMS values for one of the flights the authors examined)?*

We have cited most of the references provided here (listed below) and another relevant paper (Kim et al., 2023) and summarized the findings from those papers in Introduction. Although we believe the current method takes in the meteorological data simulated by the WRF model and does not assume the meteorological conditions were steady, the unsteady conditions still pose problems for the emission inversion. We have ackowedge this and included the following recommendation for the future flight planning, "It is also wise to choose relative steady meteorological conditions for the flight campaign since the unsteady conditions such as frequent wind direction changes pose great challenges not only to the inverse modeling but also to the meteorological simulation and the dispersion model."

However, the shortcoming pointed out by this referee applies to the mass balance method. The current method takes in the meteorological data simulated by the WRF model and does not assume the meteorological conditions were stable during the observation time.

Akingunola, A., Makar, P. A., Zhang, J., Darlington, A., Li, S.-M., Gordon, M., Moran, M. D., and Zheng, Q.: A chemical trans-port model study of plume-rise and particle size distribution for the Athabasca oil sands, Atmos. Chem. Phys., 18, 8667–8688, https://doi.org/10.5194/acp-18-8667-2018, 2018.

Fathi, S., Gordon, M., Makar, P. A., Akingunola, A., Darlington, A., Liggio, J., Hayden, K., and Li, S.-M.: Evaluating the impact of storage-and-release on aircraft-based mass-balance methodology using a regional air-quality model, Atmos. Chem. Phys., 21, 15 461–15 491, https://doi.org/10.5194/acp-21-15461-2021, 2021.

Gordon, M., Makar, P. A., Staebler, R. M., Zhang, J., Akingunola, A., Gong, W., and Li, S.-M.: A comparison of plume rise algorithms to stack plume measurements in the Athabasca oil sands, Atmos. Chem. Phys., 18, 14 695–14 714, https://doi.org/10.5194/acp-18-14695- 2018, 2018.

Liggio, J., Li, S.-M., Hayden, K., Taha, Y. M., Stroud, C., Darlington, A., Drollette, B. D., Gordon, M., Lee, P., Liu, P., Leithead, A., Moussa, S. G., Wang, D., O'Brien, J., Mittermeier, R. L., Brook, J. R., Lu, G., Staebler, R. M., Han, Y., Tokarek, T. W., Osthoff, H. D., Makar, P. A., Zhang, J., Plata, D. L., and Gentner, D. R.: Oil sands operations as a large source of secondary organic aerosols, NATURE, 534, 91+, https://doi.org/10.1038/nature17646, 2016.

Kim, J., Seo, B.-k., Lee, T., Kim, J., Kim, S., Bae, G.-N., and Lee, G.: Airborne estimation of SO2 emissions rates from a coal-fired power plant using two top-down methods: A mass balance model and Gaussian footprint approach, Sci. Total Environ., 855, https://doi.org/10.1016/j.scitotenv.2022.158826, 2023.

- *Introduction, page 2, lines 48-56: One difference between retrievals of cesium, volcanic ash, wildfire particulate emissions and unreactive tracer transport, and emissions of SO2, is that SO2 may undergo oxidation by the OH radical, as well as uptake and oxidation within cloud droplets, creating sulphuric acid and particle sulphate. The paper needs to recognize this loss process, and at least attempt to estimate its magnitude (e.g. state whether or not the observations were carried out under cloud-free conditions, hence eliminating the loss through cloud processing, and estimate the gas-phase oxidative losses via OH, preferably through an independent observation-derived estimate of OH concentrations and if not via typical OH concentrations). My expectation is that the OH loss will be a relatively minor term, but this needs to be confirmed). Another loss process that needs to be explored is via dry deposition of SO2 ( the authors mention dry deposition in terms of its Henry's Law dependance, but not how HYSPLIT makes use of different vegetation types in its calculation of deposition. See also for example Hayden et al 2021, ACP, https://acp.copernicus.org/articles/21/8377/2021/) for a description of how aircraft retrievals of SO2 may be used to estimate SO2 deposition velocities directly. Note that the accuracy of Briggs' equations may also depend on the manner in which they are implemented (see for example Gordon et al., 2018: https://acp.copernicus.org/articles/18/14695/2018/) and/or Akingunola et al, 2018:*

*https://acp.copernicus.org/articles/18/8667/2018/). The latter suggests that applying Briggs' formulae may be more accurate on a layer by layer basis to account for changes in the atmospheric temperature profile with height. The authors should provide a bit more implementation details on how they used Briggs equations: does HYSPLIT or the driving WRF meteorology include information on temperature in the vertical, hence allowing for a layer-by-layer approach instead of assuming that the surface conditions are sufficient to determine the plume height?*

Green et al. (2019) found that the SO 2 oxidation rates during the day from power plants were 0.22–0.71%/hr, using 13 flights from 6 February to 15 March 2015 over the eastern United States. The measurements were made during a clear-sky day on March 26, 2019 and the travel time of the measured air parcels from the stacks are less than three hours. So it is reasonable to treat SO 2 as a passive tracer and ignore its oxidation.

Green, J. R., Fiddler, M. N., Holloway, J. S., Fibiger, D. L., McDuffie, E. E., Campuzano-Jost, P., Schroder, J. C., Jimenez, J. L., Weinheimer, A. J., Aquino, J., Montzka, D. D., Hall, S. R., Ullmann, K., Shah, V., Jaegle, L., Thornton, J. A., Bililign, S., and Brown, S. S.: Rates of Wintertime Atmospheric SO2 Oxidation based on Aircraft Observations during Clear-Sky Conditions over the Eastern United States, J. of Geophys. Res., 124, 6630–6649, https://doi.org/10.1029/2018JD030086, 2019.

In the HYSPLIT model, the vegetation types are reflected in the land use input file. They are used in the resistance method when calculating the dry deposition. This has been clarified in the revision. In addition, three references below are added on the resistance method implementation that the HYSPLIT model closely follows.

Wesely, M. L.: Parameterization of surface resistances to gaseous dry deposition in regional-scale numerical models, Atmos. Environ., 23, 1293–1304, https://doi.org/10.1016/0004-6981(89)90153-4, 1989. Chang, J. S., Middleton, P. B., Stockwell, W. R., Binkowski, F. S., and Byun, D.: The regional acid deposition model and engineering model, in: Acidic deposition: State of science and technology, Vol I, Emissions, Atmospheric Processes, and Deposition, PB-92-100403/XAB, USA, https://www.osti.gov/biblio/5388896, 1990.

Walmsley, J. L. and Wesely, M. L.: Modification of coded parametrizations of surface resistances to gaseous dry deposition, Atmos. Environ., 555 30, 1181–1188, https://doi.org/10.1016/1352-2310(95)00403-3, 1996.

In the HYSPLIT model, stability is only based on the surface conditions. This has been clarified with the added text shown below. "Note that the stability parameter is calculated using the surface conditions of the meteorological data in the HYSPLIT model. A recent study by Akingunola et al. (2018) suggests a layered buoyancy approach that allows stability to change: with height for the Briggs plume rise calculation. However, the layered approach is not implemented in the HYSPLIT model yet."

- *Methods, section 2.1, line 90. It appears that there are sufficient wind speed and direction, temperature, etc., observations to determine the likelihood of retrieval success (e.g. see the three meteorology-based metrics used to describe conditions for accurate retrievals in Fathi et al, 2021). These checks can be carried out a priori to eliminate some flight data as being unlikely to provide good retrievals. The paper should include a brief description including a few explanatory figures with each flight explaining why the flight is likely to be a good candidate for successful SO2 emissions retrieval.*

The current inverse modeling method does not require some typical conditions (such as steady wind speed and uniform flow field) that a mass balance method requires, which is discussed by Fathi et al, 2021. This has been clarified by adding the text below in the introduction.

Fathi et al. (2021) investigated the impact of storage-and-release due to meteorological variability on mass-balance emission rate retrieval accuracy using virtual aircraft sampling of a regional chemical transport model output. The storage-and-release events contributed to the mass-balance emission estimate errors ranging from $-25\%$ to $24\%$ in their tests. Therefore inverse modeling methods using a dispersion model without assuming constant meteorological fields are expected to perform better than the mass balance method.

- *Figure 1, page 4: A general comment: I was surprised that the flight tracks were apparently a single downwind screen or wall, and did not attempt to bracket the source at multiple levels (box or oval flight around the source) or provide an upwind screen and downwind screen. A single downwind screen will fail to allow the effects of upwind sources to be removed from the retrieval, as well as preventing tests of meteorological variability over space to be carried out (the latter in turn providing information on the extent to which the steady-state requirement for successful retrievals is taking place). This is a fundamental drawback of the sampling methodology in the flights the authors are using to test their retrieval methodology.*

We agree that box or oval flights would help, although it is not required by the current inverse modeling method, which utilizes the relationship between sources and receptors provided by the dispersion model. While steady state is critical for successful retrievals

using the mass balance method, in theory it is not required by the inverse modeling method used in this study.

- *Methods, section 2.2, HYSPLIT model. The authors are using HYSPLIT as their model for retrievals. The authors need to make an argument for why HYSPLIT is an appropriate modelling platform for carrying out retrievals, as opposed to a public domain Eulerian model such as CMAQ. For example, they mention that HYSPLIT passively advects tracers in their configuration (line 100). SO2 is a reactive gas, being oxidized in both gas and aqueous reactions. The authors make no estimates of these potential oxidative losses, and don't mention whether the aircraft observations took place under clear sky conditions (i.e. whether or not aqueous removal may be likely). The potential for upwind sources of SO2 have not been included in the model. The potential for depositional losses of SO2 needs to be described in more detail – how does this depend on land use and what deposition algorithm is being used (reference). The model's ability to simulate turbulent mixing of pollutants, in addition to advective transport has not been described. Later (lines 108 to 111) the authors mention that WRF turbulence variances are used by HYSPLIT and that a fixed horizontal to vertical turbulence is imposed, and boundary layer heat and momentum fluxes are used to calculate boundary layer fluxes, but its not clear how HYSPLIT uses this information. Does the model include turbulent diffusive transport, for example? HYSPLIT has the advantage of being computationally fast – but I am not convinced based on the authors description that HYSPLIT will capture enough of the relevant physics and chemistry to be a good proxy for real atmosphere in the retrievals process. Had an Eulerian model such as CMAQ been used for the retrieval process, the vertical diffusion, upwind sources of emissions, and oxidative removal of SO2 would have been included by default. Or were all these processes included in the authors' HYSPLIT implementation? The authors need to describe them in this section if so.*

The following text has been added to justify the use the HYSPLIT model for the simulation of SO2.

Green et al. (2019) found that the $SO_2$ oxidation rates during the day from power plants were 0.22–0.71%/hr, using 13 flights from 6 February to 15 March 2015 over the eastern United States. The measurements were made during a clear-sky day on March 26, 2019 and the travel time of the measured air parcels from the stacks are less than three hours. So it is reasonable to treat $SO_2$ as a passive tracer and ignore its oxidation.

Green, J. R., Fiddler, M. N., Holloway, J. S., Fibiger, D. L., McDuffie, E. E., Campuzano-

Jost, P., Schroder, J. C., Jimenez, J. L., Weinheimer, A. J., Aquino, J., Montzka, D. D., Hall, S. R., Ullmann, K., Shah, V., Jaegle, L., Thornton, J. A., Bililign, S., and Brown, S. S.: Rates of Wintertime Atmospheric SO2 Oxidation based on Aircraft Observations during Clear-Sky Conditions over the Eastern United States, J. of Geophys. Res., 124, 6630–6649, https://doi.org/10.1029/2018JD030086, 2019.

Yes, the HYSPLIT model does include turbulent diffusive transport. The following text has been added to clarify this. "Random velocity components based on the meteorological data are added to the mean advection velocities to simulate the dispersion process." The details of the model can be found in Draxler and Hess (1997, 1998) and Stein et al. (2015).

- *Page 4, line 106: Large variations in the observed wind direction on time scales of one minute imply that an accurate SO2 emissions retrieval may not be possible (see the a priori meteorological criteria described in Fathi et al, 2021). A successful retrieval is dependent on the meteorology being relatively constant during the aircraft flight period. If this is not the case, estimates of emissions fluxes are likely to be in error (see Fathi et al 2021).*

Although the large variations in the observed wind direction make the estimation difficult, the inversion may still be carried out with the current method in which the HYSPLIT model can represent the storage-and-release demonstrated by Fathi et al., 2021 using the Global Environmental Multiscale-Modeling Air-Quality and CHemistry (GEM-MACH) model.

- *Page 5, lines 119-121: A better and slightly more recent reference for Briggs would be his 1984 book chapter: Briggs, G. A.: Plume rise and buoyancy effects, atmospheric sciences and power production, in: DOE/TIC-27601 (DE84005177), edited by: Randerson, D., TN, Technical Information Center, US Dept. of Energy, Oak Ridge, USA, 327–366, 1984.*

This reference has been added.

- *Page 7, lines 132-140: The authors mention here that the gas exit temperature of the three stacks could not be obtained, without explanation, whereas they later mention (line 244) that Continuous Emissions Monitoring System data for the stacks was available. In my experience, CEMS data usually includes stack temperatures – and while there isn't an official NEI 2019 year, the 2017 NEI is available, and all three facilities mentioned by the author are present (cf https://www.epa.gov/air-emissions-inventories/2017-national-emissions-inventory-nei-data#dataq). My point here is that*

*gas emissions temperature usually is part of CEMS, is included in USA inventory reporting, and should be available to the authors. The authors have made use of a range of estimates of QH: this may not have been necessary, if the stack temperatures and exit velocities are available, since these may be used to generate Fb values as well (see Briggs, 1984). More justification / description of the efforts made to determine stack gas exit temperatures or typical values for same from another year (and why these were not part of the CEMS records) need to be provided in the text (or alternatively, make use of those temperatures to estimate the Fb terms in the equations). They might also contact George Pouliot at the US EPA (the EPA's emissions guru) to ask for this information if its not available on-line (pouliot.george@epa.gov). The use of approximations for QH is a serious limitation of the authors' work here, and I'm not sure its necessary.*

It was our original expectation that the stack gas exit temperatures were part of the the CEMS records. However, the data are not in any public databases or publications. During our investigation, we contacted all the three power plants for the hourly exit temp erature data but could not get those information. We've contacted the EPA inventory staff, George Pouliot, as suggested by the reviewer. In our email exchanges, he stated that "the (hourly) stack exit temperature is not routinely reported to the EPA as part of the CEMS data." He kindly provided the assumed stack exit temperatures for these facilities based on the 2020 NEI while cautioning that the "the temperature can vary and is not constant". We estimated the $Q_H$ value using 2020 typical stack exit temperature and added the following paragraph in the discussion.

While the stack exit gas temperature data are not available for this study, a single constant stack exit temperature is provided for each facility in the 2020 National Emissions Inventory (NEI) (Personal communication with George Pouliot at the U.S. EPA). Using the average measured air temperature as the ambient temperature and the other CEMS data (United States Envi- ronmental Protection Agency (U.S. EPA), 2022), including hourly exit air flow rates, the morning/afternoon heat emissions are estimated as 52–59 MW/49–56 MW, 80–92 MW/76–87 MW, and 13–13 MW/12–13 MW, for Roxboro, Belews Creek, and CPI Roxboro, respectively. Note that the heat emission estimation is sensitive to the stack exit temperature which is expected to vary from hour to hour, similar to the exit air flow rates, and the $SO_2$ emissions. Nonetheless, these estimated values indicate the reasonable ranges of the heat emission. When correlation-based and RMSE-based methods agree with each other in their "optimal" heat emission for Roxboro and Belews Creek morning segments, the "optimal" heat

emissions are very close to the estimated stack heat emissions here. When the two methods disagree, the correlation-based "optimal" heat emissions of 90 MW/140 MW for CPI Roxboro morning/afternoon are unreasonably high, but the RMSE-based "optimal" emissions of 30 MW/50 MW could still be reasonable. This suggests that the RMSE-based results are probably more reliable.

- *Section 2.4: Its worth noting here the potential impact of model deficiencies on the emissions estimates generated using the cost function and HYSPLIT output (see my note above). This is a generic concern with a data assimilation approach – the accuracy of the model used may influence the accuracy of the resulting retrieved emissions. The emissions generated will be those required to create the most accurate HYSPLIT predictions – but if HYSPLIT itself does not do a good job of transport, reaction, etc., of the emissions, then the resulting emissions estimates may be inaccurate (especially if some of the physical processes known to be present in the actual atmosphere are absent in the model). This role of model physical parameterization detail and prediction accuracy on the retrieval process via data assimilation methods such as cost function minimization needs to be acknowledged in the text. The authors should also contrast with other methods of observation-based emissions estimation which do not have this limitation, but may have other limitations (e.g. see Fathi et al, 2021 for some examples and references).*

As noted earlier, we have added discussion on the limitations of the different retrieval methods in Introduction, referencing the related work by Gordon et al. (2018), Akingunola et al. (2018), Fathi et al. (2021), and Kim et al. (2023). In addition, the following paragraph is also added to the summary and discussion section to acknoledge the potential problem of the data assimilation approach used in this study.

While the uncertainty of the heat emission is focused here, there are a lot of other uncertainties associated with the emission estimates. For instance, uncertainties in many parameters, such as the assumed background $SO_2$ mixing ratios, the meteorological data input such as the wind direction and speed, and some of the HYSPLIT turbulence parameterizations related to the turbulent mixing, will all affect the final results. Even if the hourly exit temperature were available, the plume rise calcu- lated using the Briggs algorithm may still misplace the plume. It is likely that the "optimal" heat emissions chosen here have compensated other errors in the model.

- *Lines 181-183, page 8: the variation in background SO2 mixing ratio would presumably have been captured with a reaction-transport model such as CMAQ.*

While chemical transport models may have an advantage of including many reactions, they are not better at the transport simulation especially near the source when no plume-in-grid is not employed, due to numerical diffusions introduced to the grid cells in the Eulerian models.

- *Line 184, page 8: I'm used to measurement campaigns where the aircraft flights are determined based on a model forecast to limit the amount of time the aircraft samples air that is not from the sources. A few words on why there was no contribution from the powerplants before 15Z or after 21Z should be added here: was this due to the aircraft flying to/from the plumes during those times, or some other consideration?*

The following two sentences have been added.

"Apparently the $SO_2$ emitted from the power plant stacks before 15Z have been transported out of the region when the aircraft measurement were made along the flight routes. Figure 1b shows that aircraft has left the domain of interest at 21Z so that $SO_2$ emitted after 21Z were not sampled either"

- *Lines 188-189, page 8: the use of the given QH value allows separation of the three plumes – is there also evidence from the observations that the three plumes were separated (e.g. in a flight screen, you'd get three different hotspots in the wall, separated by low concentrations)? The uncertainty in the interpretation in turn suggests that getting observed stack gas temperatures is critical, see earlier comment.*

Yes, Figures A1-10 (Figure 10, and Figures A1–9 in the revision) mostly show hotspots separated by low concentrations. We agree that getting the actual hourly stack gas temperatures would be ideal. However, it is not possible to obtain such data with our best efforts.

- *Figure 5, page 11: presumably the observations can be used to estimate both plume and PBL heights – they should be included on this image. How well did the model perform relative to observations?*

Based on the vertical profiles of the potential temperature, the PBL heights are estimated as 1330 m and 1750 m AGL at around 15:23Z and 20:41Z, respectively. They have been added to Figure 5. This indicates that the model may have underestimated the PBL heights. The statement, "Figure 5 also shows that the WRF PBL heights appear to be underestimated when compared with the two observation-based PBL heights estimated using the vertical potential temperature profiles", has been added to the main text.

- *Section 3.2 a general comment: I understand the value of a sensitivity run of a model to determine its sensitivity to a key parameter (in this case QH). There is also a question regarding the accuracy of the methodology used to retrieve emissions. Here, we have a sensitivity run that suggests that the retrieval accuracy is highest for particular QH values. Can an argument be made whereby deficiencies in the model or the retrieval algorithm be ruled out as alternative causes for the retrieval to work well at the given QH value? This is why I'm hoping that typical stack gas temperatures are available; if they are used instead of set of QH sensitivity values to estimate Fb, then the estimate of QH can be ruled out as a cause of error and the focus can be on the accuracy of the retrieval and the model. My concern here is that HYSPLIT lacks sufficient process detail to do a good job of representing SO2 removal and transport, in which case the choice of QH may compensate for that lack of detail... with an impact on the estimated emissions. A more detailed description of how SO2 is modelled in the authors' HYSPLIT implementation might help to alleviate this concern. Figure 6: ok, the correlation coefficient improves depending on the choice of QH value – how do we know that the QH value is actually correct? Line 259-260: the correlation between model and obs has improved for certain QH. The authors seem to assume that this means that the given QH values are correct... what if it means that the given QH values compensate for other issues in the model, giving a correct result but for the wrong reason? Is there any additional information that can be brought to bear to indicate that the given QH values when the model performs well and hence Fb are in fact accurate? See above comments regarding EPA data, etc...*

It is possible that the optimal $Q_H$ which produces the best results may compensate other model deficiencies. If the accurate hourly stack gas exit temperatures are available, the plume rise calculated using the Briggs algorithm probably will not match the observations perfectly. However, allowing a free parameter to be adjusted likely helps to yield an optimal simulation. We also ackoweledged that the "optimal" $Q_H$ may have compensated other issues in the model (see the 5th paragraph in "Summary and disscusion").

- *Table 1, page 13: The authors show that the model performs better in morning than afternoon. Why is this the case? They mention errors in the WRF wind fields – I'm wondering about other issues, such as the wind direction and stability changing over time. Was the model stability and boundary layer height stable during the time period (see the criteria for a successful retrieval in Fathi et al, 2021 – these might provide additional explanations regarding why the model performance is poor (and why*

*retrievals during those times are less likely to be successful).*

While the errors in the WRF wind might be the main reason for the performance difference between morning and afternoon cases, the meteorological variability is also important, especially for the retrievals. We added the following paragraph at the end of Seciton 3.2.

Table 1 also hows that the model simulation generally performs better in the morning than in the afternoon. This is probably related to the fact that the wind directions in the afternoon are more variable than in the morning, as shown in Fig. 3. The meteorological variability may cause storage-and-release events which make successful emission estimation more difficult to obtain, especially for the mass balance method (Fathi et al., 2021).

- *Section 3.3.1, 3.3.2: given the impact of background SO2, it would be worthwhile to mention that the flight patterns themselves could have been better constructed, to sample upwind as well as downwind air as has been done in other studies (see above references and references quoted therein). This would have removed background SO2 levels as a source of uncertainty in the emissions estimates.*

The following two sentences have been added to the beginning of Section 3.3.2, at the end of the first paragraph.

It has to be noted that the flight patterns could have been better constructed. Sampling upwind as well as downwind or in closed shape flight patterns which enclose the sources (e.g., see Ryoo et al. (2019), Fathi et al. (2021), and Kim et al. (2023)) would have helped significantly in the estimation of the SO2 background mixing ratios.

Ryoo, J.-M., Iraci, L. T., Tanaka, T., Marrero, J. E., Yates, E. L., Fung, I., Michalak, A. M., Tadić, J., Gore, W., Bui, T. P., Dean-Day, J. M., and Chang, C. S.: Quantification of CO 2 and CH 4 emissions over Sacramento, California, based on divergence theorem using aircraft measurements, Atmos Meas Tech., 12, 2949–2966, https://doi.org/10.5194/amt-12-2949-2019, 2019.

- *Line 295, page 14: why was the height of the plumes not estimated from the aircraft observations and included in Figure 5? If the "best performance" QH values are correct, and the underlying plume rise algorithm and model are also correct, then the plume heights should match fairly well. Do they?*

We agree that the plume heights can be determined if there are enough observations to capture the vertical $SO_2$ profiles. However, our current data do not have the needed

information, as shown later in the vertical profiles of the observations. Although some photographs were taken during the morning and afternoon flights, it is still impossible to determine the plume heights based on those.

- *Section 3.3.3. The variation in model results based on correlation coefficient versus RMSE show the sensitivity of the model to the chosen QH, and that the "optimal" QH may depend on the statistic used for the evaluation. Can this comparison provide the reader with any evaluation of the accuracy of the plume rise method itself?*

We do not think the comparison provide evaluation of the accuracy of the plume rise method itself. Here the comparison is intended to show which metric is better to provide the "optimal" QH for the inversion. Based on the results, RMSE is probably a better metric for this purpose.

- *Figure 9: it is difficult to tell from the plots as presented whether the model is doing a good or poor job of simulating the plume height. It would be better if the authors aggregated the observations to allow for individual plumes to be determined (e.g. by interpolation between the observation values) and the same time aggregation applied to the measurements: that is, how well do the average plume heights for each plume resolved in the observations compare to the average plume heights determined by the model in each case?*

We agree that it is much better if we can determine the plume height from the observations. In the figure below, we aggregated the observations and the correlation-based/RMSE-based "optimal" predictions as a function of altitude for the plumes from the three power plants during the morning and afternoon flights. Due to the lack of observations at multiple heights, it is impossible to resolve the plume height with the available observations. The multiple $SO_2$ mixing ratios at a single altitude most reflect the horizontal variations when the aircraft traveled across the plume.

- *Summary/ Discussion:*

*Based on the study results, what would the authors recommend for future aircraft and emissions estimate follow-up work (e.g. flight planning to include upwind SO2 measurements, measuring sources for which CEMS data including stack parameters are available, a priori decision making for when the data is suitable for retrievals and when it is not suitable, etc..*

The following paragraph has beened added at the end.

[Figure]

Figure 1: Aggregated $SO_2$ observations and the correlation-based/RMSE-based "optimal" predictions as a function of altitude for the plumes from the three power plants during the morning (upper row) and afternoon (lower row) flights.

This study shows that RMSE is a better metric than correlation coefficient in choosing the best ensemble member for the $SO_2$ emission inversion. While the RMSE-based "optimal" plume rise runs appear to agree better with the observations than the correlation-based "optimal" runs, observations are often missing when and where the "optimal" runs are significantly different. Additional measurements at mulitple altitudes would have been really helpful. In the future flight planning of similar top-down emission estimation studies more vertical profiles of the target pollutant should be measured. In addition, more upwind measurements are also recommended in order to better quantify the background concentrations caused by many other emission sources. It is also wise to choose relative steady meteorological conditions for the flight campaign since the unsteady conditions such as frequent wind direction changes pose great challenges not only to the inverse modeling but also to the meteorological models and the dispersion modeling. The current study shows the value of the ensemble simulations when certain model parameters are difficult to determine, such as stack heat emission here.

- *Is the use of RMSE better than correlation coefficient in determining emissions? I'm not clear on that by the end of the paper.*

  RMSE appears to be better than correlation coefficient in providing the "optimal" emission estimation. It has been made clear in the revision. For instance, the last sentences in the abstract now read "The RMSE-based "optimal" runs result in a much better agreement with the CEMS data for the previous severely overestimated segment and do not deterioirate much for the other segments, with relative errors as 18%, -18%, 3%, -9%, and 27% for the five segments, and 2% for Belews Creek afternoon segment. In addition, the RMSE-based "optimal" heat emissions appear to be more reasonable than the correlation-based values when they are significantly different for CPI Roxboro power plant."

- *Line 431-435: the authors need to include in this discussion the need for direct observations of plume gas temperatures – which are included in most large point source observations (I'm hoping they can get this data from the web links and email contact address I've included above). The authors apparently believe that plume observations do not include stack temperature observations – I've found the contrary to be the case in my experience, looking at emissions inventories. It would be unusual for the power-plants mentioned to not also have stack parameters, so some follow-up by the authors is worthwhile.*

  We've contacted the EPA inventory staff, George Pouliot, as suggested by the reviewer. In our email exchanges, he stated that "the (hourly) stack exit temperature is not routinely reported to the EPA as part of the CEMS data." While a single constant stack exit temperature is given in the annual NEI, it is definitely not a constant number and can vary from hour to hour. During our investigation, we did contact all the three power plants for the hourly exit temperature data but could not get those information.

- *Lines 441-443: "We speculate. . . " please clarify and expand on this statement. Do you mean plume placement in the vertical dimension? And why would the method be less sensitive to a height error?*

  Here it is meant to be the horizontal placement. This has been clarified. Clearly the different vertical placements in the vertical dimension using different $Q_H$ yields different emission estimates. So the method is quite sensitive to the height error.